# Brain-imaging evidence for compression of binary sound sequences in human memory

**Fosca Al Roumi[1]\*[†], Samuel Planton[1†], Liping Wang[2], Stanislas Dehaene[1,3]**

[1]Cognitive Neuroimaging Unit, Université Paris-Saclay, INSERM, CEA, CNRS, NeuroSpin center, Gif/Yvette, France; [2]Institute of Neuroscience, Key Laboratory of Primate Neurobiology, CAS Center for Excellence in Brain Science and Intelligence Technology, Chinese Academy of Sciences, Shanghai, China; [3]Collège de France, Université Paris Sciences Lettres (PSL), Paris, France

**Abstract** According to the language-of-thought hypothesis, regular sequences are compressed in human memory using recursive loops akin to a mental program that predicts future items. We tested this theory by probing memory for 16-item sequences made of two sounds. We recorded brain activity with functional MRI and magneto-encephalography (MEG) while participants listened to a hierarchy of sequences of variable complexity, whose minimal description required transition probabilities, chunking, or nested structures. Occasional deviant sounds probed the participants' knowledge of the sequence. We predicted that task difficulty and brain activity would be proportional to the complexity derived from the minimal description length in our formal language. Furthermore, activity should increase with complexity for learned sequences, and decrease with complexity for deviants. These predictions were upheld in both fMRI and MEG, indicating that sequence predictions are highly dependent on sequence structure and become weaker and delayed as complexity increases. The proposed language recruited bilateral superior temporal, precentral, anterior intraparietal, and cerebellar cortices. These regions overlapped extensively with a localizer for mathematical calculation, and much less with spoken or written language processing. We propose that these areas collectively encode regular sequences as repetitions with variations and their recursive composition into nested structures.

**\*For correspondence:**
fosca.alroumi@cea.fr

[†]These authors contributed equally to this work

**Competing interest:** The authors declare that no competing interests exist.

## Editor's evaluation

This article brings to bear an important set of behavioral methods and neural data reporting that activity in numerous cortical regions robustly covaries with the complexity of tone sequences encoded in memory. The study provides convincing evidence that humans store these sequences based on representations related to the so-called language of thought.

## Introduction

The ability to learn and manipulate serially ordered lists of elements, that is sequence processing, is central to several human activities (**Lashley, 1951**). This capacity is inherent to the ordered series of subtasks that make up the actions of daily life, but is especially decisive for the implementation of high-level human skills such as language, mathematics, or music. In non-human primates, multiple levels of sequence encoding ability, with increasing complexity, have been identified, from the mere representation of transition probabilities and timings to ordinal knowledge (which element comes first, second, third, etc.), recurring chunks, and even abstract patterns (e.g. does the sequence obey

the pattern xxxxY, i.e. a repetition ending with a different element) (*Dehaene et al., 2015*; *Jiang et al., 2018*; *Shima et al., 2007*; *Wang et al., 2015*; *Wilson et al., 2013*). We and others, however, proposed that the representation of sequences in humans may be unique in its ability to encode recursively nested hierarchical structures, similar to the nested phrase structures that linguists postulate to underlie human language (*Dehaene et al., 2015*; *Fitch and Martins, 2014*; *Hauser et al., 2002*). Building on this idea, it was suggested that humans would spontaneously encode temporal sequences of stimuli using a language-like system of nested rules, a 'language of thought' (LOT; *Fodor, 1975*; *Al Roumi et al., 2021*; *Amalric and Dehaene, 2017a*; *Chater and Vitányi, 2003*; *Feldman, 2000*; *Li and Vitányi, 1993*; *Mathy and Feldman, 2012*; *Planton et al., 2021*; *Wang et al., 2019*). For instance, when faced with a sequence such as xxYYxYxY, humans may encode it using an abstract internal expression equivalent to '2 groups of 2, and then an alternation of 4'.

The assumption that humans encode sequences in a recursive, language-like manner, was recently tested with a non-linguistic visuo-spatial task, by asking human adults and children to memorize and track geometric sequences of locations on the vertices of an octagon (*Al Roumi et al., 2021*; *Amalric et al., 2017b*; *Wang et al., 2019*). Behavioral and brain-imaging studies showed that such sequences are internally compressed in human memory using an abstract 'language of geometry' that captures their numerical and geometrical regularities (e.g. 'next element clockwise', 'vertical symmetry'). Indeed, behavioral results showed that the difficulty of memorizing a sequence was linearly modulated, not by the actual sequence length, but by the length of the program capable of generating it using the proposed language ('minimal description length' [MDL]; for a definition and brief review, see *Dehaene et al., 2022*). The length of this program provides a predictor of sequence complexity. We will, from now on, refer to it as LoT complexity. In a follow-up fMRI experiment where participants had to follow the same sequences with their gaze, activity in the dorsal part of inferior prefrontal cortex correlated with the LoT complexity while the right dorsolateral prefrontal cortex encoded the presence of embedded structure (*Wang et al., 2019*). These results indicate that sequences are stored in memory in a compressed manner, the size of this code being the length of the shortest program that describes the sequence in the proposed formal language. Memory for sequences would therefore follow the 'MDL' principle inherited from information theory (*Grunwald, 2004*) and often used to capture various human behaviors (*Chater and Vitányi, 2003*; *Feldman, 2000*; *Mathy and Feldman, 2012*). *Wang et al., 2019*, further showed that the encoding and compression of such sequences involved brain areas supporting the processing of mathematical expressions rather than language-related areas, suggesting that multiple internal languages, not necessarily involving classical language areas, are present in the human brain. In a follow-up study, *Al Roumi et al., 2021*, showed with MEG that the spatial, ordinal, and geometrical primitive codes postulated in the proposed LoT could be extracted from brain activity.

In the present work, we ask whether this LoT may also explain the human memory for binary auditory sequences (i.e. sequences made up of only two possible items, for instance two sounds with high and low pitch, respectively). While arguably minimal, binary sequences preserve the possibility of forming structures at different hierarchical levels. They therefore provide an elementary window into the mental representation of nested language-like rules, and which aspect of this representation, if any, is unique to the human species. While it would make little sense to ask if non-human animals can store spoken human sentences, it does seem more reasonable to submit them to a protocol with minimal, binary sound sequences, and ask whether they use a recursive language-like format for encoding in memory, or whether they are confined to statistical learning or chunking. The latter mechanisms are important to consider because they are thought to underpin the processing of several aspects of sequence processing in human infants and adults as well as several animal species, such as the extraction of chunks within a stream of syllables, tones, or shapes (*Fló et al., 2019*; *Santolin and Saffran, 2018*; *Toro and Trobalón, 2005*; *Saffran et al., 1996*), or the community structure that generates a sequence of events (*Karuza et al., 2019*; *Schapiro et al., 2013*). Yet, very few studies have tried to separate the brain mechanisms underlying rule-based predictions from those of probabilistic sequence learning (*Bhanji et al., 2010*; *Kóbor et al., 2018*; *Maheu et al., 2021*). Our goal here is to develop such a paradigm in humans, and to test the hypothesis that human internal models are based on a recursive LoT.

The present work capitalized on a series of behavioral experiments (*Planton et al., 2021*) in which we proved that human performance in memorizing binary auditory sequences, as tested by the

capacity to detect occasional violations, could be predicted by a modified version of the language of geometry, based on the hierarchical combination of very few primitives (repeat, alternate, concatenate, and integers). This work considered binary sequences of various lengths (from 6 to 16 items) mainly in the auditory but also in the visual modality, and showed that LoT complexity was correlated with participants' oddball detection performance. This was especially true for longer sequences of 16 items as their length exceeds typical working memory capacity (*Cowan, 2001*; *Cowan, 2010*; *Miller, 1956*) and therefore requires compression. In this work, LoT predictions were compared to competitor models of cognitive complexity and information compression (*Aksentijevic and Gibson, 2012*; *Alexander and Carey, 1968*; *Delahaye and Zenil, 2012*; *Gauvrit et al., 2014*; *Glanzer and Clark,*

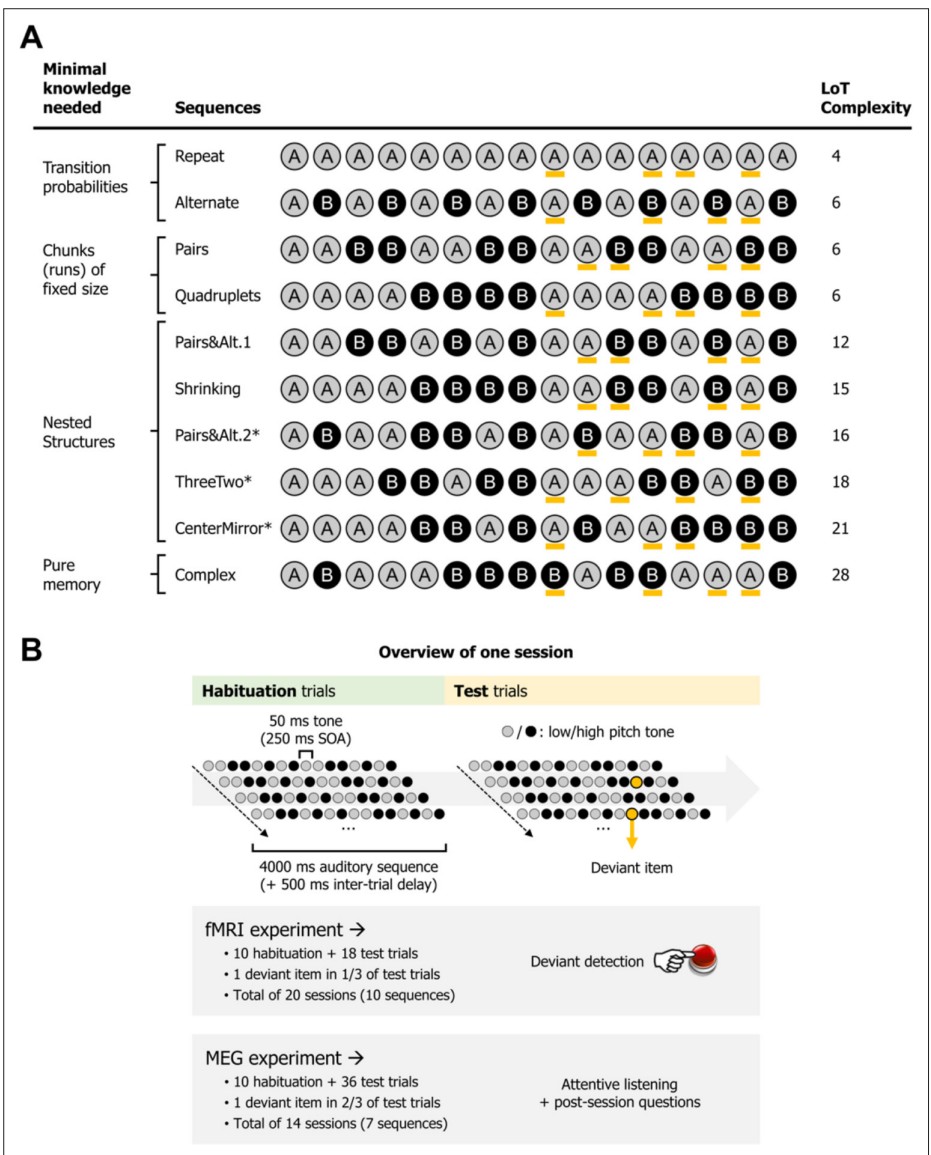

**Figure 1.** Experimental design. (**A**) List of the different 16-item sequences used in the magneto-encephalography (MEG) and fMRI experiments, with associated language of thought (LoT) complexity, and categorized according to the type of knowledge assumed to be required for optimal memory encoding. Orange marks indicate positions in which violations could occur (4 possible positions per sequence, all between positions 9 and 15). *Sequences used only in the fMRI experiment. Sequence description provided by the LoT and the corresponding verbal description are provided in *Supplementary file 1*. (**B**) Overview of the presentation paradigm (example with a session the Pairs&Alt.1 sequence), with the respective characteristics of the fMRI experiment and the MEG experiment. One unique sequence was used in any given session. Each sequence was tested twice, in two different sessions (reversing the mapping between A/B items and low/high pitch).

*1963*; *Psotka, 1975*; *Vitz and Todd, 1969*). The predictive power of LoT outperformed all competing theories (*Planton et al., 2021*).

Here, we use functional MRI and magneto-encephalography (MEG) to investigate the cerebral underpinnings of the proposed language in the human brain. We exposed participants to 16-item auditory binary sequences, with varying levels of regularity, while recording their brain activity with fMRI and MEG in two separate experiments (see *Figure 1*). By combining these two techniques, we aimed at obtaining both the spatial and the temporal resolution needed to characterize in depth the neural mechanisms supporting sequence encoding and compression.

In both fMRI and MEG, the experiment was composed of two phases. In a habituation phase, the sequences were repeatedly presented in order for participants to memorize them, thus probing the complexity of their internal model. In a test phase, sequences were occasionally presented with deviants (a single tone A replacing another tone B), thus probing the violations of expectations generated by the internal model (*Figure 1B*). We focused on a very simple prediction arising from the hierarchical predictive coding framework (*Friston, 2005*). According to this view, and to much experimental research (e.g. *Bekinschtein et al., 2009*; *Chao et al., 2018*; *Heilbron and Chait, 2018*; *Summerfield and de Lange, 2014*; *Wacongne et al., 2011*), the internal model of the sequence, as described by the postulated LoT, would be encoded by prefrontal regions of the brain, which would send anticipation signals to auditory areas, where they would be subtracted from incoming signals. As a consequence, we predict a reciprocal effect of LoT on the brain signals during habituation and during deviancy. In the habituation part of the experiment, lower amplitude response signals should be observed for sequences of low complexity – and conversely, during low complexity sequences, we expect top-down predictions to be stronger and therefore deviants to elicit larger responses, than for complex, hard to predict sequences.

Two subtleties further qualify this overall theoretical picture. First, at the highest level of sequence complexity, sequences cannot be compressed in a simple LoT expression, and therefore we expect the brain areas involved in nested sequence coding to exhibit no further increase in activation, or even a decrease (*Vogel and Machizawa, 2004*; *Wang et al., 2019*). We will evaluate the presence of such a non-linear trend by testing a quadratic term for LoT complexity in addition to a purely linear term in the regression models. Second, a simpler system of statistical learning, based on transition probabilities, may operate in parallel with LoT predictions (*Bekinschtein et al., 2009*; *Chao et al., 2018*; *Maheu et al., 2019*; *Maheu et al., 2021*; *Meyniel et al., 2016*; *Summerfield and de Lange, 2014*). To separate their contributions, we will use multiple predictors in a general linear model (GLM) of behavior and of MEG signals, whose temporal resolution allows to track individual sequence items (in fMRI, the BOLD response was too slow relative to the sequence rate of 250 ms per item).

## Results

### The LoT for binary sequences

As used in the present work, the LoT hypothesis postulates that humans encode mental objects such as sequences or geometric shapes using 'mental programs', that is expressions in a symbolic language characterized by (1) a small set of primitives and (2) the capacity to recursively combine these primitives through three operators: concatenation, repetition (possibly with variations), and recursive nesting (i.e. calling of a subprogram) (*Dehaene et al., 2022*; *Sablé-Meyer et al., 2022*). In this framework, defining a language requires the selection of a vocabulary of primitives that be recursively combined.

The language for binary sequences that we evaluate here is an adaptation of a LoT for spatial sequences with geometrical regularities, which accounted for participants' predictions in an explicit sequence completion task (*Amalric and Dehaene, 2017a*) and in a gaze anticipation task (*Wang et al., 2019*). Previous fMRI and MEG studies have found neural evidence that human participants use such an LoT to encode geometrical sequences in memory and predict upcoming items (*Al Roumi et al., 2021*; *Wang et al., 2019*). For an in-depth description of this language, please see supporting information in *Amalric et al., 2017b*.

In *Planton et al., 2021*, we showed how the very same language could also account for the compression of auditory sequences made of two sounds. To obtain this language for binary sequences, instead of the eight vertices of the octagon, we consider only two states, A and B (e.g. a high-pitch and a low-pitch tone). The primitive operations are now reduced to the *stay* operation (called *+0*) and the

*change* operation (called *b*). Note that, similarly to a Turing machine, the descriptions are sequential: each operation is relative to the previous state. The repetition operation, denoted by ^n, allows any sequence of operations to be repeated *n* times, possibly with variations, denoted by <>. For instance, the instruction '[xxx]^2< +0 >' indicates that the expression [xxx] is executed twice, with the same starting point: '< +0 >', while '[xxx]^2< b >' indicates that the expression [xxx] is executed twice, with a change in the starting point: '< b >' (see *Supplementary file 1*).

The description length of a mental program is computed as a weighted sum of the fixed costs attached to each primitive instruction (see supporting information in *Amalric et al., 2017b*). The cost for repeating an instruction *n* times is assumed to correspond to log10(*n*) (rounded up), that is the number of digits needed to encode the number in decimal notation. Note that any given sequence has several possible descriptions. For instance, AAAA could be described as [+0,+0,+0,+0] or in a more compact manner as [+0]^4. LoT complexity is the MDL of a sequence, that is the shortest possible description of it in the proposed language.

In our previous work, we compared extensively the proposed LoT to other sequence encoding models such as entropy, change complexity, or algorithmic complexity (*Planton et al., 2021*). The findings indicated that LoT complexity for chunk-preserving expressions, that is those that do not split any chunk of repetitions, provided the best fit of participants' behavior, over and above all other competing models. The present study builds on these results and investigates the neural code of the LoT.

## Stimulus design

We designed a hierarchy of sequences (*Figure 1*) of fixed length (16 items) that should systematically vary in complexity according to our previously proposed LoT (*Planton et al., 2021*) and whose gradations separate the lower-level representations of sequences that may be accessible to non-human primates (as outlined in *Dehaene et al., 2015*) from the more abstract ones that may only be accessible to humans (*Figure 1A*).

*Figure 1* presents the sequences we selected for the experiments and their complexit. In *Supplementary file 1*, we also give the formal LoT description and a short verbal description of their minimal program. The sequences formed a hierarchy. At the lowest level, much evidence indicates that the brain spontaneously encodes statistical regularities such as transition probabilities in sequential sensory inputs and uses them to make predictions (e.g. *Barascud et al., 2016*; *Bendixen et al., 2009*; *McDermott et al., 2013*; *Meyniel et al., 2016*; *Saffran et al., 1996*), an ability well within the grasp of various non-human animals (e.g *Hauser et al., 2001*; *Meyer and Olson, 2011*). The first two sequences in our hierarchy therefore consisted in predictable repetitions (AAAA…) and alternations (ABABA…). In terms of information compression, such sequences can be represented with a very short LoT expression (a mere repetition, or a repetition of alternations).

At the next level, we tested chunking, the ability to group a recurring set of contiguous items into a single unit, another major sequence encoding mechanism which is also accessible to non-human primates (*Buiatti et al., 2009*; *Fujii and Graybiel, 2003*; *Saffran et al., 1996*; *Sakai et al., 2003*; *Uhrig et al., 2014*). Thus, we included sequences made of pairs (AABBAABB…) or quadruplets (AAAABBBB…). Our LoT model attributes them a high level of compressibility, but already some degree of hierarchy (a loop of chunks). Relative to the previous sequences, they require monitoring the number of repetitions before a new chunk starts (ABABA…=1; AABBAA…=2; AAAABBBB…=4), and may therefore be expected to engage the number system, though to involve the bilateral intra-parietal sulci, particularly their horizontal and anterior segments (*Dehaene et al., 2003*; *Eger et al., 2009*; *Harvey et al., 2013*; *Kanayet et al., 2018*).

The next level required nested structures, that is a hierarchical representation of smaller chunks embedded in larger chunks. Although there is some debate on whether this level of representation can be accessed by non-human animals, especially with extensive training (*Ferrigno et al., 2020*; *Gentner et al., 2006*; *Jiang et al., 2018*; *van Heijningen et al., 2009*), many agree that the ability to access it quickly and spontaneously is a potential human-specific trait in sequence learning and many related cognitive domains (*Dehaene et al., 2015*; *Fitch and Hauser, 2004*; *Fitch and Martins, 2014*; *Hauser et al., 2002*). We probe it using a variety of complex but compressible sequences such as 'AABBABABAABBABAB' (whose hierarchical description is $[A^2B^2[AB]^2]^2$ and can be paraphrased as '2 pairs then 4 alternations, repeated twice'). Here again, our LoT model easily compresses such nested

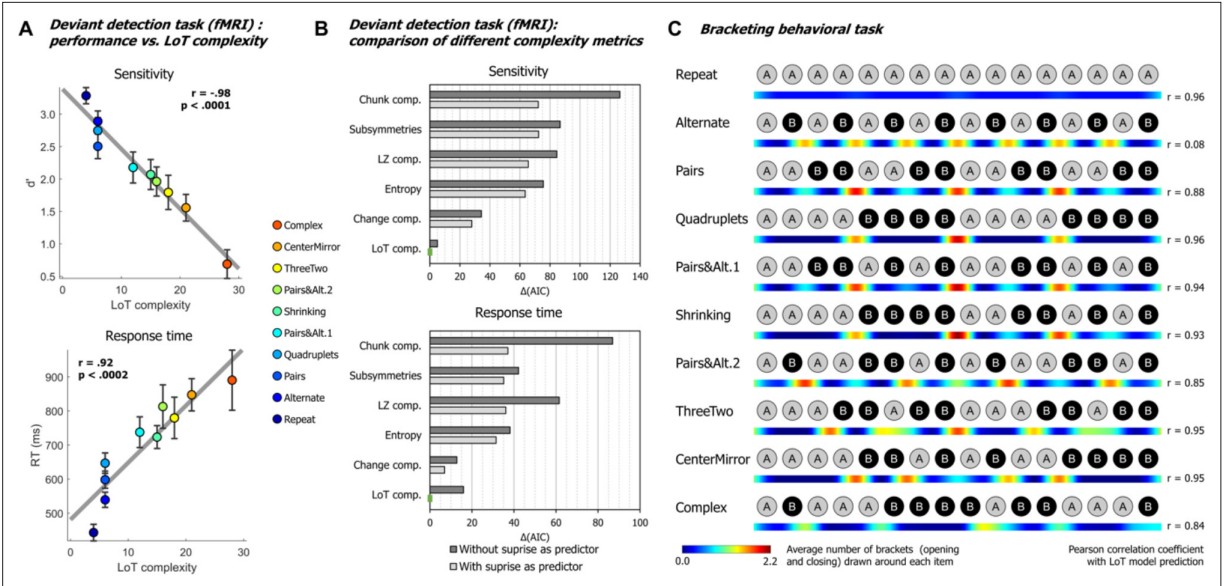

**Figure 2.** Behavioral data. (**A**) Group-averaged sensitivity (*d′*) and response times for each sequence in the deviant detection task, plotted against the language of thought (LoT) complexity. A significant linear relationship with LoT complexity was found in both cases. The Pearson correlation coefficient and associated p-value are reported. Error bars represent SEM. (**B**) Comparison of the goodness of fit (indexed by the Akaike information criterion) of 12 mixed models (for sensitivity, top, and for response time, bottom), that is, each testing one out of six different complexity metrics (see main text) and including or not a transition-based surprise predictor. Δ(AIC) is the difference in AIC with the best model of the 12. A lower value indicates a better fit. The best model (Δ(AIC)=0) is marked by a green rectangle on the vertical axis. (**C**) The heatmap for each sequence represents the vector of the average number of brackets drawn by the participants around each item in the sequence bracketing task (after smoothing for illustration purposes). The Pearson correlation coefficient with the vector of brackets predicted by the LoT model is reported on the right side. A high correlation was obtained for all sequences but *Alternate*, since several subjects segmented this sequence in eight groups of two items, while the shortest LoT expression represented it in a single group of 16 items with 15 alternations.

The online version of this article includes the following figure supplement(s) for figure 2:

**Figure supplement 1.** Task performance: average sensitivity (*d′*), for each position and each sequence.

structures by using only one additional bit whenever a chunk needs to be repeated, regardless of its hierarchical depth (for details, see *Amalric and Dehaene, 2017a*).

Finally, as a control, our paradigm also includes a high-complexity, largely incompressible sequence, with balanced transition probabilities and minimal chunking possibilities. We selected a sequence which our language predicted to be of maximal complexity (highest MDL), and which was therefore predicted to challenge the limits of working memory (*Figure 1A*). Note that because such a complex sequence, devoid of recurring regularities, is not easily encodable within our language (except for a trivial concatenation of chunks), we expect the brain areas involved in nested sequence coding to exhibit no further increase in activation, or even a decrease (*Vogel and Machizawa, 2004*).

## Behavior in deviant detection

Similar to *Planton et al., 2021*, we used task performance in the fMRI deviant detection task to quantify the LoT model's ability to predict behavior. Sensitivity (*d′*) was calculated by examining the hit rate for each sequence and each violation position, relative to the overall false-alarm rate on standard no-violation trials. On average, participants managed to detect the deviants at above chance level in all sequences and at all positions (min *d′*=0.556, min t(22) = 2.919, p<0.0080). Thus, they were able to detect a great variety of violation types in regular sequences (unexpected alternations, repetitions, change in number, or chunk boundaries). However, performance worsened as the 16-item sequences became too complex to be easily memorized. The group-averaged performance in violation detection for each sequence (regardless of deviant position) was linearly correlated with LoT complexity, both for response times to correctly detected items (RTs) (F(1, 8)=43.87, p<0.0002, *R²*=0.85) and for sensitivity (*d′*) (F(1, 8)=159.4, p<0.0001, *R²*=0.95) (see *Figure 2A*). When including the participants as a random factor in a linear mixed

model, we obtained a very similar result for sensitivity (F(1, 206)=192.92, p<0.0001, with estimates of –0.092±0.007 for the LoT complexity predictor, and 3.39±0.17 for the intercept), as well as for responses times (F(1, 203)=110.87, p<0.0001, with estimates of 17.4 ms±1.6 for the LoT complexity predictor, and 475.4 ms±38.6 for the intercept). As for false alarms, they were rare, and no significant linear relationship with LoT complexity was found in group averages (F(1, 8)=2.18, p=0.18), although a small effect was found in a linear mixed model with participant as the random factor (F(1, 206)=4.83, p<0.03, with estimates of 0.038±0.017 for the LoT complexity predictor, and 1.57±0.39 for the intercept).

We evaluated whether these results could be explained by statistical learning, that is whether deviants were more easily or more rapidly detected when they violated the transition probabilities of the current sequence. For sensitivity (*d'*), a likelihood ratio test showed that adding a transition-probability measure of surprise (*Maheu et al., 2019*; *Meyniel et al., 2016*) to the linear regression with LoT complexity improved the goodness of fit ($\chi^2(1)$=4.35, p<0.038). The effect of transition-based surprise was indeed significant in the new model (F(1, 205)=4.35, p<0.039), but LoT complexity effect remained highly significant (F(1, 205)=106.28, p<0.0001). Similarly, for RTs, adding transition-based surprise to the model significantly improved model fit ($\chi^2(1)$=12.27, p<0.0005). Transition-based surprise explained some of the variance in RTs (F(1, 202)=12.51, p<0.0006), but again the effect of LoT complexity remained highly significant (F(1, 202)=46.3, p<0.0001).

We also added a quadratic complexity term to the models to determine whether the trend was purely linear (i.e., performance degrading continuously with complexity) or also had a non-linear component (e.g. 'plateau' performance after reaching a certain level of complexity). For sensitivity, a quadratic term did not improve goodness of fit, whether transition-based surprise was also included ($\chi^2(1)$=1.58, p=0.21) or not ($\chi^2(1)$=0.02, p=0.89). For RTs, it did not improve goodness of fit in the model including transition-based surprise ($\chi^2(1)$=1.43, p=0.23), but it did when transition-based surprise was not included ($\chi^2(1)$=6.76, p<0.010). These results indicate that the effect of complexity on behavior is primarily linear, although a non-linear trend may be present on response times, as suggested by examining *Figure 2A*.

We then assessed whether alternative models of sequence complexity could better explain the present behavioral data, in an analysis similar to *Planton et al., 2021*. The five alternative accounts we considered were: *Entropy*, a measure of information that quantifies the uncertainty of a distribution, here based on transition pairs (*Maheu et al., 2021*); *Lempel-Ziv complexity*, derived from the popular lossless data compression algorithm (*Lempel and Ziv, 1976*); *number of subsymmetries*, proposed by *Alexander and Carey, 1968*, which is the number of symmetric subsequences of any length within a sequence; *chunk complexity*, a measure of the number of runs or chunks weighted by their length (*Mathy and Feldman, 2012*; *Planton et al., 2021*); and *change complexity*, a metric proposed by *Aksentijevic and Gibson, 2012*, quantifying the average amount of 'change' across all subsequences contained in a sequence (see *Planton et al., 2021*, for additional details on these metrics). As before, for each of these metrics plus LoT, we fitted two linear mixed models, with and without the transition-based surprise regressor, resulting in 12 models for sensitivity and 12 models for RTs. All models included participants as a random factor along with the fixed factor(s). The Akaike information criterion (AIC) was used as an indicator of goodness of fit. As presented in *Figure 2B*, for sensitivity, the best models to predict performance were, in this order, 'LoT complexity + surprise' (AIC = 582.9, conditional $R^2$=0.62), 'LoT complexity' (AIC = 587.9, conditional $R^2$=0.62) and 'change complexity + surprise' (AIC = 610.8, conditional $R^2$=0.55). For RTs, the best models were 'LoT complexity + surprise' (AIC = 3031.2, conditional $R^2$=0.55), 'change complexity + surprise' (AIC = 3038.1,, conditional $R^2$=0.52) and 'change complexity' (AIC = 610.8, conditional $R^2$=0.52). Regardless of the complexity metric, adding the transition-based surprise regressor in the model always resulted in improved goodness of fit (reduction in AIC of 18.54 on average for sensitivity, and 18.49 for RTs).

In summary, using a partially different set of sequences, we replicated the behavioral findings of *Planton et al., 2021*: for long sequences that largely exceed the storage capacity in working memory, violation detection and response speed (both indexing the ease of memorizing the sequence and anticipating the next item) were well correlated with the LoT model of sequence compression, which outperformed other approaches for quantifying sequence complexity.

## Behavioral bracketing task

After brain imaging, we also asked all participants to report their intuitions of how each sequence should be parsed by drawing brackets on a visual representation of its contents (after listening to it). The results (see heatmaps in *Figure 2C*) indicated that participants agreed about how a sequence should be parsed and used bracketing levels appropriately for nested sequences. For instance, they consistently placed brackets in the middle of sequences that consisted of two repetitions of eight items, but did so less frequently both within those phrases and when the midpoint was not a predicted parsing point (CenterMirror in *Figure 2C*). In order to assess the correspondence between the parsings and the organization proposed by the LoT model, we computed for each sequence the correlation between the group-averaged number-of-brackets vector and the LoT model vector (obtained from the sequence segmentation derived from the LoT description in terms of repeat, alternate, and concatenate instructions). A strong correlation was found for sequences *Repeat* (Pearson $r=0.96$, $p<0.0001$), *Pairs* ($r=0.88$, $p<0.0001$), *Quadruplets* ($r=0.96$, $p<0.0001$), *Pairs&Alt.1* ($r=0.94$, $p<0.0001$), *Shrinking* ($r=0.93$, $p<0.0001$), *Pairs&Alt.2* ($r=0.85$, $p<0.0001$), *ThreeTwo* ($r=0.95$, $p<0.0001$), *Center-Mirror* ($r=0.95$, $p<0.0001$), and *Complex* ($r=0.84$, $p<0.0001$), but not for *Alternate* ($r=0.08$, $p=0.77$). For the latter, a departure from the proposed encoding was found: while the shortest LoT representation encodes it as '15 alternations', the participants' parses corresponded to '8 AB pairs'. In the discussion, we explain how this small departure from theory could have arisen from the specifics of the visual bracketing task, rather than the actual encoding of the auditory sequence.

It could be suggested that, rather than the structure predicted by the LoT model, participants use transition probabilities to segment a sequence, with rare transitions acting as chunk boundaries. We thus tested the correlations between the group-averaged number-of-brackets vector and the transition-based surprise derived from transition probabilities. There are 15 item-to-item transitions in 16-item sequences, thus brackets before the first and after the last items were excluded from this analysis. Surprise was computed by pooling over all the transitions in a given sequence. Due to lack of variance, a correlation with transition-based surprise was impossible for sequences *Repeat* (all transitions are 100% predictable repetitions) and *Alternate* (all transitions are 100% predictable alternations). For other sequences, a positive correlation was found for sequences *Quadruplets* ($r=0.99$, $p<0.0001$), *Pairs* ($r=0.88$, $p<0.0001$), *Complex* ($r=0.79$, $p<0.0005$), *Shrinking* ($r=0.73$, $p<0.002$), *ThreeTwo* ($r=0.68$, $p<0.006$), and *CenterMirror* ($r=0.64$, $p<0.02$). Crucially, however, no positive correlation was found for sequences *Pairs&Alt.1* ($r=–0.48$, $p=0.071$) and *Pairs&Alt.2* ($r=–0.58$, $p<0.03$). This was due to the fact that repetitions were the rarest and therefore the most surprising transitions in these sequences: thus transition probabilities predicted a breaking of the chunks of repeated item, while participants correctly did not do so and placed their brackets at chunk boundaries (*Planton et al., 2021*). Therefore, although surprise arising from the learning of transition probabilities can partially predict participants' bracketing behavior in some cases, notably when a sequence is composed of frequent repetitions and rare alternations, this model completely fails in other (e.g. when repetitions are rare), again indicating that higher-level models such as LoT are needed to explain behavior.

In summary, using a partially different set of sequences, we replicated the behavioral findings of *Planton et al., 2021*, showing that, especially for long sequences that largely exceed the storage capacity in working memory, violation detection (an index of learning quality) and response speed (potentially indexing the degree of predictability) were well predicted by our LoT model of sequence compression.

## fMRI data

### A positive effect of complexity during sequence learning and tracking

As predicted, during the habituation phase (i.e. during sequence learning), fMRI activation mostly increased with sequence complexity in a broad and bilateral network involving supplementary motor area (SMA), precentral gyrus (preCG) abutting the dorsal part of Brodmann area 44, cerebellum (lobules VI and VIII), superior and middle temporal gyri (STG/MTG), and the anterior intraparietal sulcus (IPS) region (close to its junction with the postcentral gyrus) (*Figure 3A* and *Table 1*). These regions partially overlapped with those observed in sequence learning for a completely different domain, yet a similar language: the visuo-spatial sequences of *Wang et al., 2019*. In the opposite direction, a reduction of activation with complexity was seen in a smaller network, mostly corresponding to the default-mode network, which was increasingly deactivated as working memory load

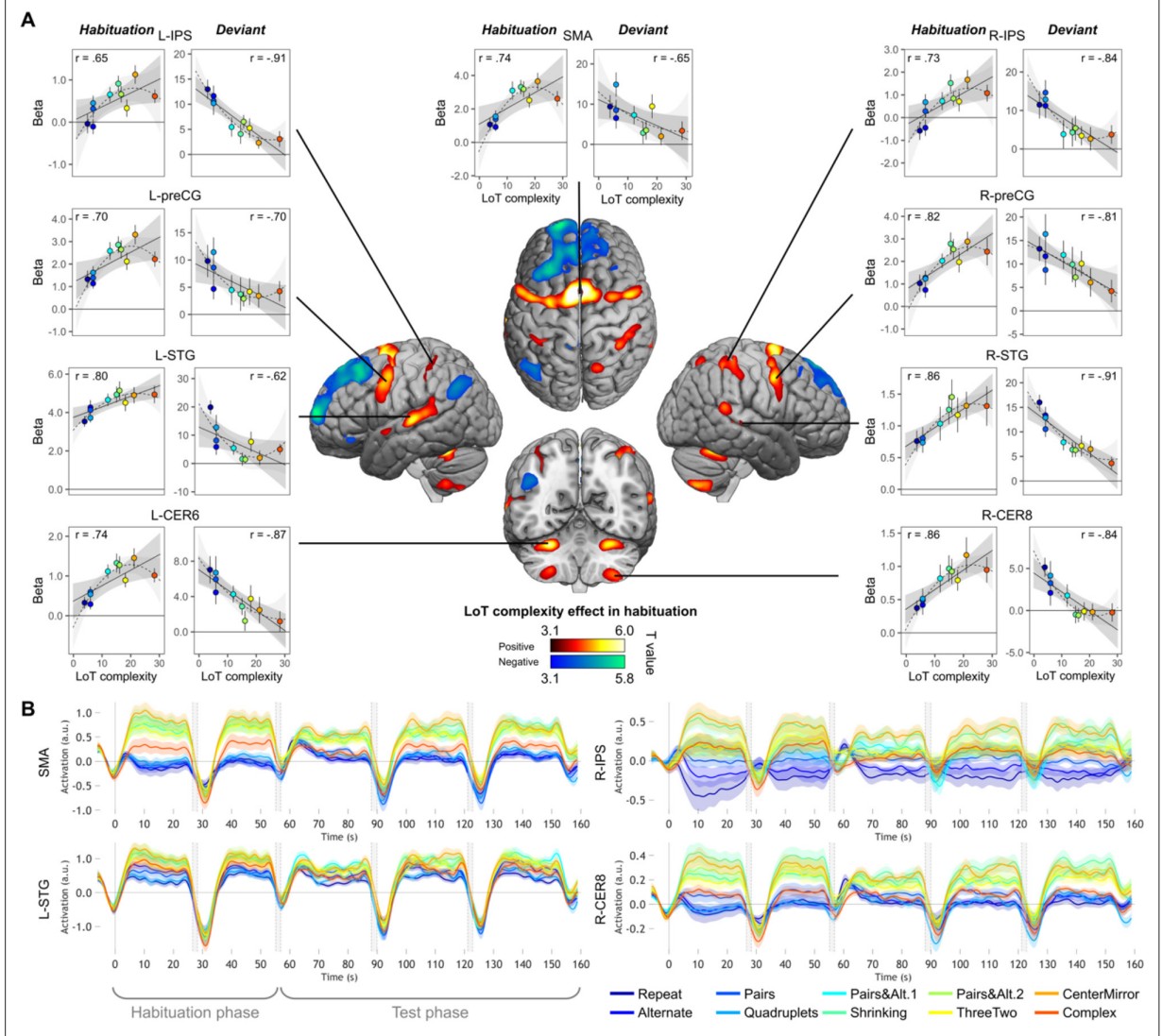

**Figure 3.** Sequence complexity in the proposed language of thought (LoT) modulates fMRI responses. (**A**) Brain areas showing an increase (hot colors) or a decrease (cold colors) in activation with sequence LoT complexity during habituation (voxel-wise p<0.001, uncorrected; cluster-wise p<0.05, FDR corrected). Scatterplots represent the group-averaged activation for each of the ten sequences as a function of their LoT complexity (left panels: habituation trials; right panels, deviant trials) in each of nine regions of interest (ROIs). Data values are from a participant-specific ROI analysis. Error bars represent SEM. Linear trends are represented by a solid line (with 95% CI in dark gray) and quadratic trend by a dashed line (with 95% CI in light gray). Pearson linear correlation coefficients are also reported. (**B**) Time course of group-averaged BOLD signals for each sequence, for four representative ROIs. Each mini-session lasted 160 s and consisted of 28 trials divided into 5 blocks (2×5 habituation and 3×6 test trials), interspersed with short rest periods of variable duration (depicted in light gray). The full time course was reconstituted by resynchronizing the data at the onset of each successive block (see Materials and methods). Shading around each time course represents one SEM.

The online version of this article includes the following figure supplement(s) for figure 3:

**Figure supplement 1.** Positive (hot colors) and negative (cold colors) effects of language of thought (LoT) complexity effects on standard trials (voxel-wise p<0.001, uncorrected; cluster-wise p<0.05, FDR corrected).

**Figure supplement 2.** Time course of group-averaged BOLD signals for each sequence in nine regions of interest (ROIs) where a language of thought (LoT) complexity effect was found.

increased (**Mazoyer et al., 2001**; **Raichle, 2015**): medial frontal cortex, left middle cingulate gyrus, left angular gyrus (AG), and left pars orbitalis of the inferior frontal gyrus (IFGorb) (**Table 1**).

We then computed the same contrast with the standard trials of the test phase (sequences without violation). The network of areas showing a positive complexity effect was much smaller than during habituation: it included bilateral superior parietal cortex extending into the precuneus, left dorsal

**Table 1.** Coordinates of brain areas modulated by language of thought (LoT) complexity during habituation.

*Positive LoT complexity effect in habituation trials*

| Region | H | k | p(unc.) | p(FWE-corr) | T | x | y | z |
|---|---|---|---|---|---|---|---|---|
| | L/R | 8991 | <0.0001 | <0.0001 | 6.62 | 1 | 5 | 65 |
| | | | <0.0001 | <0.001 | 5.82 | 8 | 12 | 49 |
| Supplementary motor area, precentral gyrus, superior frontal gyrus (dorsolateral), middle frontal gyrus | | | <0.0001 | <0.05 | 5.59 | 27 | 5 | 52 |
| Lobule VIII of cerebellar hemisphere | L | 1411 | <0.0001 | <0.0001 | 6.19 | 22 | 68 | 53 |
| Lobule VI and Crus I of cerebellar hemisphere | L | 939 | <0.0001 | <0.001 | 5.97 | 29 | 56 | 28 |
| | L | 2022 | <0.0001 | <0.05 | 5.56 | 68 | 23 | 5 |
| | | | <0.0001 | <0.05 | 4.80 | 59 | 35 | 12 |
| Superior temporal gyrus, middle temporal gyrus | | | <0.0001 | 0.213 | 4.25 | 55 | 42 | 23 |
| Lobule VI of cerebellar hemisphere | R | 1216 | <0.0001 | <0.05 | 5.45 | 27 | 58 | 27 |
| | L | 1549 | <0.0001 | <0.05 | 5.04 | 22 | 67 | 53 |
| Lobule VIII of cerebellar hemisphere | | | <0.0001 | 0.118 | 4.44 | 33 | 54 | 55 |
| | R | 1039 | <0.0001 | <0.05 | 4.93 | 48 | 30 | 3 |
| | | | <0.0001 | <0.05 | 4.79 | 67 | 44 | 17 |
| Superior temporal gyrus | | | <0.001 | 0.880 | 3.55 | 69 | 23 | 3 |
| | R | 1478 | <0.0001 | <0.05 | 4.79 | 36 | 46 | 56 |
| | | | <0.0001 | 0.061 | 4.63 | 46 | 35 | 61 |
| Postcentral gyrus, Inferior parietal gyrus | | | <0.0001 | 0.170 | 4.33 | 46 | 32 | 47 |
| | R | 547 | <0.0001 | 0.085 | 4.54 | 17 | 67 | 58 |
| Superior parietal gyrus, Precuneus | | | <0.001 | 0.792 | 3.65 | 24 | 60 | 42 |
| | L | 1570 | <0.0001 | 0.106 | 4.47 | 31 | 42 | 44 |
| | | | <0.0001 | 0.149 | 4.37 | 45 | 35 | 38 |
| Inferior parietal gyrus, Postcentral gyrus | | | <0.0001 | 0.530 | 3.90 | 40 | 42 | 61 |

*Negative LoT complexity effect in habituation trials*

| Region | H | k | p(unc.) | p(FWE-corr) | T | x | y | z |
|---|---|---|---|---|---|---|---|---|
| | L/R | 12366 | <0.0001 | <0.001 | 5.86 | 19 | 67 | 12 |
| | | | <0.0001 | <0.05 | 5.42 | 29 | 25 | 47 |
| Superior frontal gyrus (dorsolateral, medial, medial orbital), middle frontal gyrus | | | <0.0001 | <0.05 | 5.33 | 6 | 44 | 58 |
| Middle cingulate and paracingulate gyri, precuneus | L | 1444 | <0.0001 | <0.05 | 5.26 | 1 | 33 | 51 |
| | L | 1530 | <0.0001 | 0.060 | 4.63 | 43 | 65 | 37 |
| | | | <0.001 | 0.816 | 3.63 | 33 | 54 | 24 |
| Angular gyrus | | | <0.001 | 0.938 | 3.45 | 27 | 82 | 44 |
| | L | 522 | <0.0001 | 0.354 | 4.07 | 52 | 35 | 14 |
| | | | <0.0001 | 0.473 | 3.95 | 34 | 40 | 7 |
| IFG pars orbitalis | | | <0.0001 | 0.645 | 3.80 | 27 | 33 | 16 |

premotor area, as well as two cerebellar regions (right lobule IV, left lobule VIII) (*Figure 2—figure supplement 1*, *Supplementary file 2*). These areas were also found during the habituation phase, although the (predominantly left) parietal superior/precuneus activation was larger and extended more posteriorly than during habituation. Regions showing a negative LoT complexity effect in standard trials (reduced activation for increasing complexity) were more numerous: medial frontal regions,

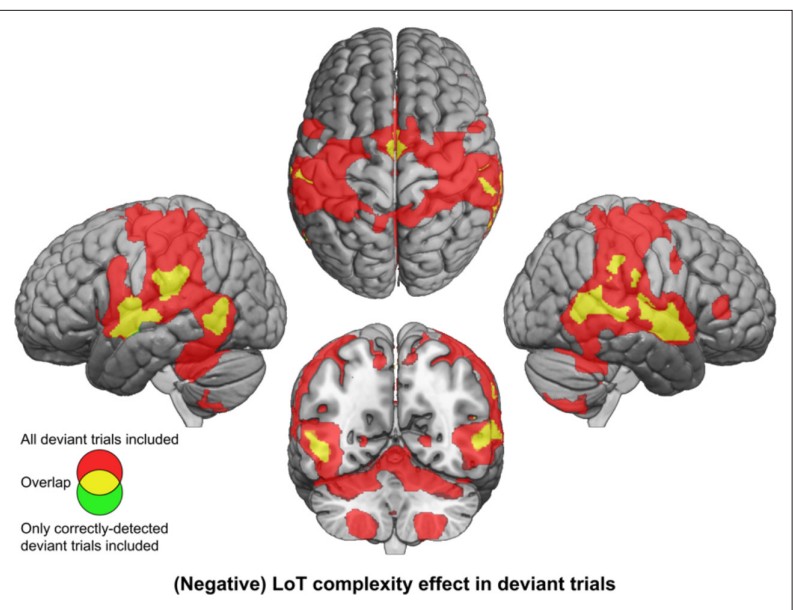

**Figure 4.** Brain responses to deviants decrease with language of thought (LoT) complexity. Colors indicate the brain areas whose activation on deviant trials decreased significantly with complexity, in two distinct general linear models (GLMs): one in which all deviant stimuli were modeled (red), and one in which only correctly detected deviant stimuli were modeled (green) (voxel-wise p<0.001, uncorrected; cluster-wise p<0.05, FDR corrected). Overlap is shown in yellow.

middle cingulate gyri, bilateral AG, bilateral anterior part of the inferior temporal gyrus, bilateral putamen, as well as left frontal orbital region and left occipital gyrus. Here again, they largely resemble what was already observed in habituation trials (i.e. a deactivation of a default-mode network), with a few additional elements such as the putamen.

## A negative effect of complexity on deviant responses

The effect of LoT complexity at the whole-brain level was first assessed on the responses to all deviant stimuli (whether detected or not). A positive linear effect of complexity was only found in a small cluster of the medial part of the superior frontal gyrus (SFG) (*Supplementary file 3*). As predicted, a much larger network showed a negative effect (i.e. reduced activation with complexity or increased activation for less complex sequences): bilateral postcentral gyrus (with major peak in the ventral part), supramarginal gyrus (SMG), IPS, STG, posterior MTG, ventral preCG, insula, SMA and middle cingulate gyrus, cerebellum (lobules VI, VIII, and vermis), right pars triangularis of the IFG (red activation map of *Figure 4*, *Supplementary file 3*). This network is thus the possible brain counterpart of the increase in deviant detection performance observed as sequences become less and less complex. However, this result could be partly due to a motor effect, since manual motor responses to deviants were less frequent for more complex sequences, as attested by an effect of LoT complexity on sensitivity. We therefore computed the same contrast using an alternative GLM modeling-only deviant trials to which the participant correctly responded (note that this model consequently included fewer trials, especially for higher complexity sequences). Negative effects of LoT complexity were still present in this alternative model, now unconfounded by motor responses. As shown in *Figure 4* (yellow) the negative effect network was a subpart of the network identified in the previous model, and concerned bilateral STG, MTG, SMG/postcentral gyrus, insula, SMA, and middle cingulate gyrus. A positive effect was still present in a medial SFG cluster, part of the default-mode network showing less deactivation for deviants as complexity increased.

## ROI analyses of the profile of the complexity effect

We next used individual ROIs to measure the precise shape of the complexity effect and test the hypothesis that (1) activation increases with complexity but may reach a plateau or decrease for the

most complex, incompressible sequence; and (2) on deviant trials, the complexity effect occurs in the opposite direction. Because merely plotting the signal at peaks identified by a linear or quadratic contrast would bias the results (**Vul et al., 2009**), we designed cross-validated individual ROI analyses, which consisted in (1) considering half of the runs to identify responsive subject-specific voxels within each ROI, using a contrast of positive effect of complexity during habituation; and (2) considering the other half to extract the activation levels for each standard or deviant sequence. Only the initial search volumes were defined on the basis of the entire data from the present study, which is likely to introduce only a minimal degree of circularity. We focused on nine areas that exhibited a positive complexity contrast in habituation (**Figure 3**), where the effect was robust and was computed on the learning phase of the experiment, therefore uncontaminated by deviant stimuli and manual motor responses.

In each ROI, mixed-effect models with participants as the random factor were used to assess the replicability of the linear effect of complexity during habituation. A significant effect was found in all ROIs (after Bonferroni correction for nine ROIs), although with variable effect size: SMA: $\beta$ estimate = 0.10, t(21) = 5.37, p.corr <0.0003; L-STG: $\beta$=0.06, t(21) = 4.75, p.corr <0.001; L-CER6: $\beta$=0.04, t(21) = 4.73, p.corr <0.002; R-IPS: $\beta$=0.07, t(21) = 4.08, p.corr <0.005; L-preCG: $\beta$=0.07, t(21) = 3.98, p.corr <0.007; R-preCG: $\beta$=0.08, t(21) = 3.8, p.corr <0.01; R-STG: $\beta$=0.03, t(21) = 3.35, p.corr <0.03; R-CER8: $\beta$=0.03, t(21) = 3.32, p.corr <0.03, and L-IPS: $\beta$=0.03, t(21) = 3.25, p.corr <0.04. These results are illustrated in **Figure 3A**, showing the linear regression trend with values averaged per condition across participants. The addition of a quadratic term was significant for seven ROIs (SMA, L-CER6, L-IPS, L-preCG, L-STG, R-CER8, and R-IPS), but did not reached significance in R-preCG nor in R-STG. This effect was always negative, indicating that the activation increases with complexity reached saturation or decreased from the most complex sequence (see dashed lines in the scatter plots of **Figure 3A**).

We also examined the time course of activation profiles within each mini-session of the experiment, that is two habituation blocks followed by three test blocks. As shown in **Figure 3B** (see **Figure 3—figure supplement 1** for all nine ROIs), the activation time courses showed a brief activation to sequences, presumably corresponding to a brief search period. 5–10 s following the first block onset, however, activation quickly dropped to a similar and very low activation, or even a deactivation below the rest level, selectively for the four lowest-complexity sequences which involved only simple processes of transition probabilities or chunking. For other sequences, the BOLD effect increased in proportion to complexity, yet with a midlevel amplitude for the most complex sequence reflecting the saturation and the quadratic effect noted earlier. Thus, in 5–10 s, the profile of the complexity effect was firmly established, and it remained sustained over time during habituation and, with reduced amplitude, during test blocks. This finding indicated that the same areas were responsible for discovering the sequence profile and for monitoring it for violations during the test period. The profile was similar across regions, with one exception: while most areas showed the same, low activation to the first four, simplest sequences, the left and right IPS showed an increasing activation as a function of the number of items in a chunk (ABABAB…=1; AABBAA…=2; AAAABBBB…=4). This observation fits with the hypothesis that these regions are involved in numerosity representation, and may therefore implement the repetitions postulated in our language.

The ROI analyses were next performed with data from the deviant trials, in order to test whether areas previously identified as sensitive to sequence complexity when learning the sequence also showed an opposite modulation of their response to deviant trials. All ROIs indeed showed a significant negative effect of LoT complexity: R-STG: $\beta$=–0.46, t(197) = –8.01, p.corr <0.0001; L-IPS: $\beta$=–0.44, t(197) = –6.35, p.corr <0.0001; R-CER8: $\beta$=–0.23, t(197) = –5.09, p.corr <0.0001; L-STG: $\beta$=–0.45, t(197) = –4.61, p.corr <0.0001; L-CER6: $\beta$=–0.23, t(197) = –4.57, p.corr <0.0001; R-preCG: $\beta$=–0.37, t(197) = –3.92, p.corr <0.002; R-IPS: $\beta$=–0.49, t(197) = –3.85, p.corr <0.002; SMA: $\beta$=–0.33, t(197) = –3.57, p.corr <0.004, and L-preCG: $\beta$=–0.27, t(197) = –2.9, p.corr <0.04. Interestingly, unlike during habituation, the addition of a quadratic term did not improve the regression except in a single area, L-STG: $\beta$=0.05, t(196) = 3.9, p.corr <0.002. Smaller effects of the quadratic term were present in three other areas, but they were not significant after Bonferroni correction: R-CER8: $\beta$=0.02, t(196) = 2.68, p<0.009; R-STG: $\beta$=0.02, t(196) = 2.42, p<0.02, and L-IPS: $\beta$=0.02, t(196) = 2.3, p<0.03.

As in the whole-brain analysis, we finally conducted a complementary analysis using activation computed with correctly detected deviants trials only. The linear LoT complexity was now only

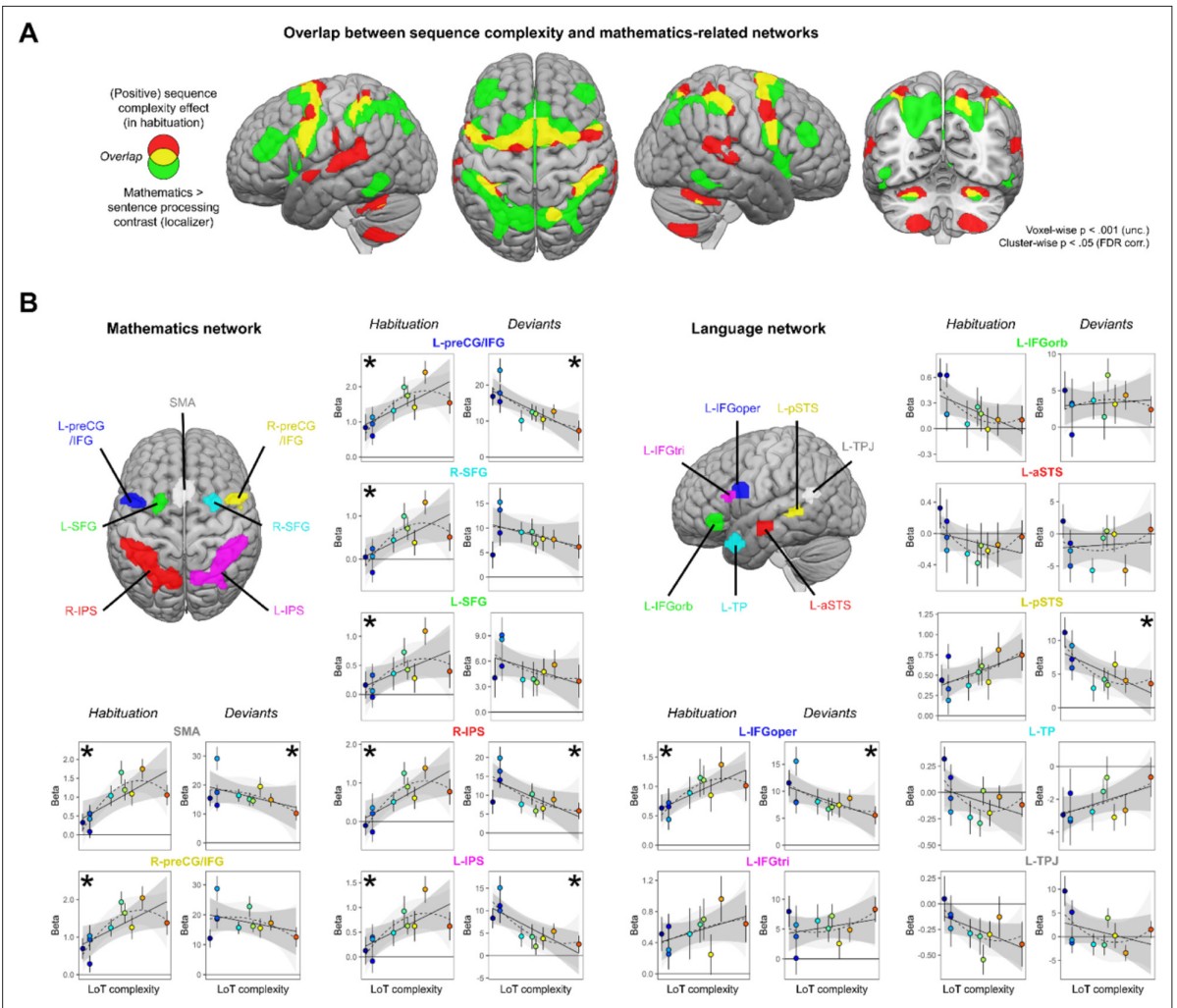

**Figure 5.** Sequence complexity effects in mathematics and language networks. (**A**) Overlap between the brain areas showing an increase of activation with sequence language of thought (LoT) complexity during habituation in the main experiment (in red) and the brain areas showing an increased activation for mathematical processing (relative to simple listening/reading of non-mathematical sentences) in the localizer experiment (in green; both maps thresholded at voxel-wise p<0.001 uncorrected, cluster-wise p<0.05, FDR corrected). Overlap between the two activation maps is shown in yellow. (**B**) Overview of the seven search volumes representing the mathematics network (left) and the seven search volumes representing the language network (right) used in the region-of-interest (ROI) analyses. Within each ROI, each scatter plot represents the group-averaged activation for each of the 10 sequences according to their LoT complexity, for habitation blocks and for deviant trials (same format as **Figure 3**). A star (*) indicates significance of the linear effect of LoT complexity in a linear mixed-effects model.

significant in four of the nine ROIs: R-STG: $\beta$=–0.48, t(197) = –7.64, p.corr <0.0001; L-IPS: $\beta$=–0.34, t(197) = –4.54, p.corr <0.0001; L-STG: $\beta$=–0.42, t(197) = –4.12, p.corr <0.0006; R-CER8: $\beta$=–0.15, t(197) = –3.17, p.corr <0.02. When adding a quadratic term, no significant effects were observed at the predefined threshold, although uncorrected ones were present for L-STG: $\beta$=0.03, t(196) = 2.37, p<0.02 and R-CER8: $\beta$=0.01, t(196) = 2.05, p<0.05.

## Overlap with the brain networks for language and mathematics

Past and present behavioral results suggest that an inner 'language' is required to explain human memory for auditory sequences – but is this language similar to natural language, or to the language of mathematics, and more specifically geometry, from which it is derived (**Al Roumi et al., 2021**; **Amalric et al., 2017b**; **Wang et al., 2019**)? By including in our fMRI protocol an independent language and mathematics localizer experiment, we tested whether the very same cortical sites are involved in natural sentence processing, mathematical processing, and auditory sequences.

At the whole-brain group level, large amount of overlap was found between the mathematics network (whole-brain mental computation > sentences processing contrast, in a second-level ANOVA of the localizer experiment) and the LoT complexity network (see *Figure 5A*): SMA, bilateral precentral cortex, bilateral anterior IPS, and bilateral cerebellum (lobules VI). Some overlap was also present, to a lower extent, with the language network (auditory and visual sentences > auditory and visual control stimuli) and the LoT complexity network: left STG, SMA, left precentral gyrus, and right cerebellum.

Such group-level overlap, however, could be misleading since they involve a significant degree of intersubject smoothing and averaging. For a more precise assessment of overlap, we extracted, for each subject and within each of seven language-related and seven math-related ROIs (see *Figure 5*), the subject-specific voxels that responded, respectively, to sentence processing and to mental calculation (same contrasts as above, but now within each subject). We then extracted the results from those ROIs and examined their variation with LoT complexity in the main experiment (during habituation). In the language network, a significant positive effect of LoT complexity during the habituation phase was only found in left IFGoper: $\beta$=0.03, t(197) = 4.25, p.corr <0.0005 (*Figure 5B*). In fact, most other language areas showed either no activation or were deactivated (e.g. IFGorb, anterior superior temporal sulcus [aSTS], temporal pole [TP], temporoparietal junction [TPJ]). As concerns deviants, a significant negative effect of LoT complexity was found in left IFGoper: $\beta$=–0.23, t(197) = –3.04, p.corr <0.04; and in left posterior superior temporal sulcus (pSTS): $\beta$=–0.24, t(197) = –3.27, p.corr <0.02. The quadratic term was never found significant.

On the contrary, in the mathematics-related network, all areas showed a positive LoT complexity effect in habituation (*Figure 5B*): SMA: $\beta$=0.05, t(197) = 5.6, p.corr <0.0001; left preCG/IFG: $\beta$=0.05, t(197) = 5.03, p.corr <0.0001; right IPS: $\beta$=0.05, t(197) = 4.69, p.corr <0.0001; right preCG/IFG: $\beta$=0.05, t(197) = 4.56, p.corr <0.0002; right SFG: $\beta$=0.04, t(197) = 4, p.corr <0.002; left IPS: $\beta$=0.04, t(197) = 3.78, p.corr <0.003 and left SFG: $\beta$=0.02, t(197) = 3.15, p.corr <0.03. The quadratic term in the second model was also significant for three of them: SMA: $\beta$=–0.01, t(196) = –4.11, p.corr <0.0009; right preCG/IFG: $\beta$=0, t(196) = –3.21, p.corr <0.03 and left preCG/IFG: $\beta$=0, t(196) = –3.1, p.corr <0.04. A negative complexity effect for deviant trials reached significance in four areas: left IPS: $\beta$=–0.42, t(197) = –4.44, p.corr <0.0003; left preCG/IFG: $\beta$=–0.48, t(197) = –4.31, p.corr <0.0004; right IPS: $\beta$=–0.41, t(197) = –4, p.corr <0.002, and SMA: $\beta$=–0.29, t(197) = –2.97, p.corr <0.05. Their response pattern was not significantly quadratic.

To summarize, all dorsal regions previously identified as involved in mathematical-processing regions were sensitive to the complexity of our auditory binary sequences, as manifested by an increase, up to a certain level of complexity, during habituation; and, for most regions, a reduction of the novelty to deviants (especially for SMA, left preCG, and IPS). Such a sensitivity to complexity was conspicuously absent from language areas, except for the left pars opercularis of the IFG.

## MEG results

The low temporal resolution of fMRI did not permit us to track the brain response to each of the 16 successive sequence items, nor to any local sequence properties such as item-by-item variations in surprise. To address this limit, a similar paradigm was tested with MEG. To maximize signal-to-noise, especially on rare deviant trials, only seven sequences were selected (*Figures 1 and 6*). Unlike the fMRI experiment, during MEG we merely asked participants to listen carefully to the presented sequences of sounds, without providing any button response, thus yielding pure measures of violation detection uncontaminated by the need to respond.

### Neural signatures of complexity at the univariate level

We first determine if a summary measure of brain activity, the global field power, is modulated by sequence complexity. To do so, we consider the brain responses to sounds occurring in the *habituation* phase, to non-deviant sounds occurring in the test phase (referred to as *standard* sounds) and to *deviant* sounds. On *habituation* trials, the late part of the auditory response (108–208 ms) correlated positively with complexity (p=0.00024, see shaded area in the top panel of *Figure 6A*): the more complex the sequence, the larger the brain response. On *standard* trials, this modulation of the GFP by complexity had vanished (middle panel of *Figure 6A*). Finally, as predicted, the GFP computed on the *deviant* exhibited the reversed effect, that is a negative correlation with complexity on the

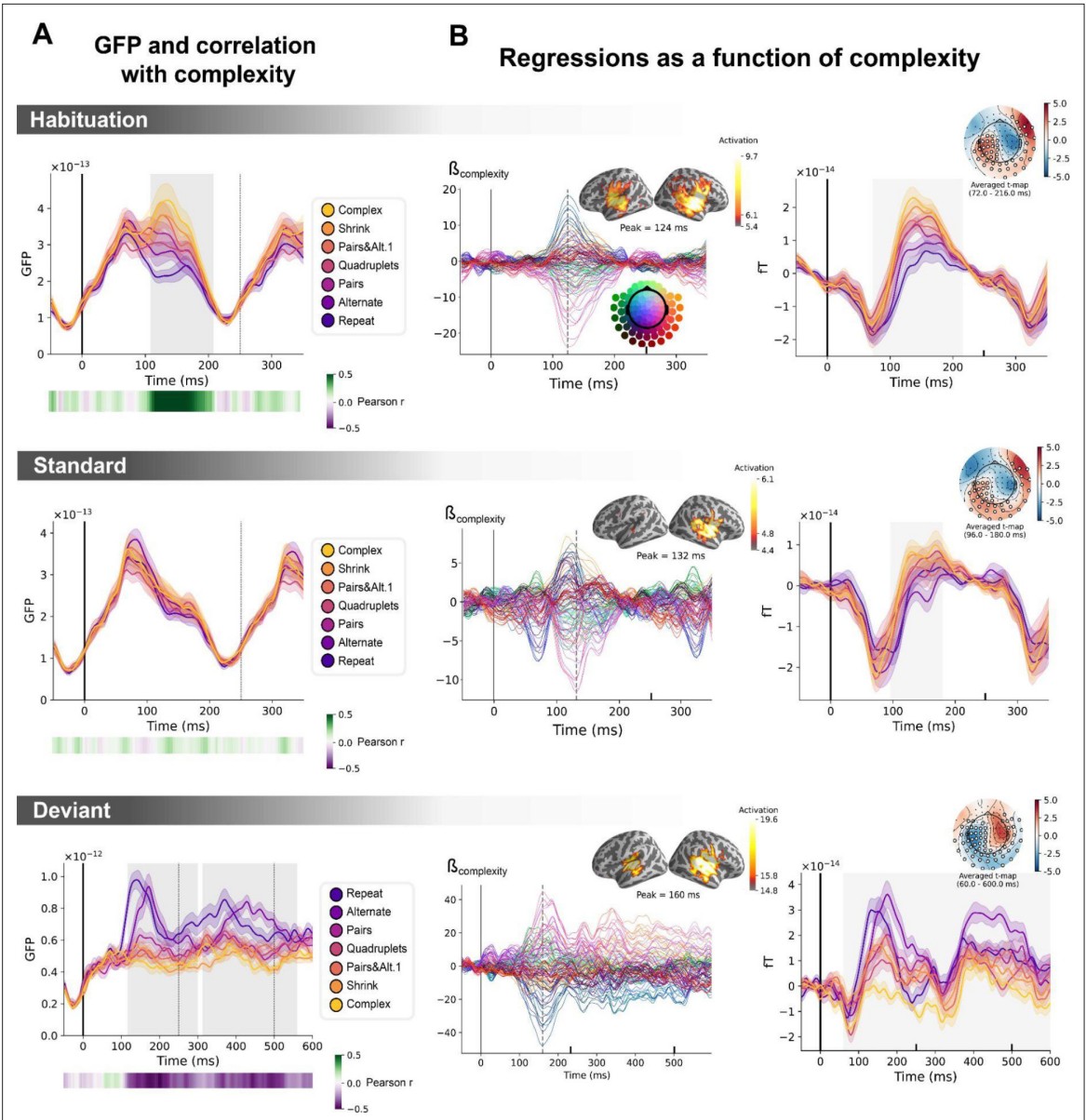

**Figure 6.** Sequence complexity in the proposed language of thought (LoT) modulates magneto-encephalography (MEG) signals to habituation, standard, and deviant trials. (**A**) Global field power computed for each sequence (see color legend) from the evoked potentials of the *habituation, standard,* and *deviant* trials. 0 ms indicates sound onset. Note that the time window ranges until 350 ms for *habituation* and *standard* trials (with a new sound onset at SOA=250 ms), and until 600 ms for *deviant* trials and for the others. Significant correlation with sequence complexity was found in *habituation* and *deviant* GFPs and are indicated by the shaded areas. (**B**) Regressions of MEG signals as a function of sequence complexity. Left: amplitude of the regression coefficients β of the complexity regressor for each MEG sensor. Insets show the projection of those coefficients in source space at the maximal amplitude peak, indicated by a vertical dotted line. Right: spatiotemporal clusters where regression coefficients were significantly different from 0. While several clusters were found (see text and *Figure 6—figure supplement 3*), for the sake of illustration, only one is shown for each trial type. The clusters involved the same sensors but on different time windows (indicated by the shaded areas) and with an opposite t-value for *deviant* trials. Neural signals were averaged over significant sensors for each sequence type and were plotted separately.

The online version of this article includes the following figure supplement(s) for figure 6:

**Figure supplement 1.** Sequence complexity modulates the contrast of deviant / matched standard trials.

**Figure supplement 2.** Unconfounding the effects of statistical surprise and sequence complexity on magneto-encephalography (MEG) signals.

**Figure supplement 3.** Spatiotemporal clusters for the complexity regressor in sensor space, shown separately for the three trial types (*habituation, standard, deviant*) and three general linear models of magneto-encephalography (MEG) signals: with complexity alone (left column); with complexity,

*Figure 6 continued on next page*

*Figure 6 continued*

transition-based surprise and repeat/alternate (middle column); and with complexity after regressing out transition-based surprise and repeat/alternate signals.

**Figure supplement 4.** Amplitude of the regression coefficient β for each magneto-encephalography (MEG) sensor for the four regressors of transition statistics: repetition/alternation for item *n* (presented at *t*=0 ms), repetition/alternation for item *n*+1 (presented at *t*=250 ms), transition-based surprise for item *n,* and transition-based surprise for item *n*+1.

116–300 ms time window (p=0.0005) and on the 312–560 ms time window (p=0.0005), indicating that *deviants* elicit larger brain responses in sequences with lower complexity (bottom panel of *Figure 6A*).

To better characterize the mechanisms of sequence coding, we ran a linear regression of the evoked responses to sounds as a function of sequence complexity. Regression coefficients of the sequence complexity predictor were projected to source space. The results showed that complexity effects were present in temporal and precentral regions of the cortex. To assess the significance of the regression coefficients, we ran a spatiotemporal cluster-based permutation test at the sensor level. Several significant clusters were found for each of the three trial types (*habituation:* cluster 1 from 72 to 216 ms, p=0.0004, cluster 2 from 96 to 212 ms, p=0.0002; *standard*: cluster 1 from 96 to 180 ms, p=0.0038, cluster 2 from 96 to 184 ms, p=0.001; *deviant*: cluster 1 from 60 to 600 ms, p=0.0002, cluster 2 from 56 to 600 ms, p=0.0002). *Figure 6* illustrates one significant cluster for each trial type. See *Figure 6—figure supplement 3* for all the clusters.

The same analyses were performed on the contrast of deviants-matched standards conditions. Matched-standard trials are selected such that they matched the deviants' ordinal position, which was specific to each sequence. These results are reported in *Figure 6—figure supplement 1*. The clusters shown involve the same sensors but exhibit opposite regression signs for the brain responses to *deviant* sounds, suggesting that, as in fMRI, the same brain regions are involved in the processing of standard and deviant items but are affected by complexity in an opposite manner.

## Controlling for local transition probabilities

Several studies have shown that human EEG/MEG responses are sensitive to the statistics of sounds and sound transitions in a sequence (*Maheu et al., 2019*; *Meyniel et al., 2016*; *Näätänen et al., 1989*; *Todorovic et al., 2011*; *Todorovic and de Lange, 2012*; *Wacongne et al., 2012*), including in infants (*Saffran et al., 1996*). When listening to probabilistic binary sequences of sounds, early brain responses reflect simple statistics such as item frequency while later brain responses reflect more complex, longer-term inferences (*Maheu et al., 2019*). Since local transition-based surprise and global complexity were partially correlated in our sequences, could this surprise alone account for our results? To disentangle the contributions of transition probabilities and sequence structure in the present brain responses, we regressed the brain signals as a function of complexity and of surprise based on transition probabilities. To capture the latter, we added several predictors: the presence of a repetition or an alternation and the surprise of an ideal observer that makes optimal inferences about transition probabilities from the past 100 items (see *Maheu et al., 2019*, for details). Both predictors were computed for two consecutive items: the one at stimulus onset (*t*=0 ms) and the next item (*t*=250 ms later) and included together with LoT complexity as multiple regressors of every time point.

*Figure 6—figure supplement 2* shows the temporal profile of the regression coefficient for sequence complexity for each MEG sensor and its projection onto the source space, once these controlling variables were introduced. The contribution of auditory regions was slightly diminished compared to the simple regression of brain signals as a function of complexity. To assess the significance of the regression coefficient, we ran a spatiotemporal cluster-based permutation test at the sensor level. Several significant clusters were found for each of the three trial types (*habituation:* cluster 1 from 96 to 244 ms, p=0.0162, cluster 2 from 112 to 220 ms, p=0.014; *standard*: cluster 1 from 104.0 to 180.0 ms, cluster value=1.50, p=0.0226, cluster 2 from 100 to 220 ms, p=0.0004; *deviant*: cluster 1 from 224 to 600 ms, p=0.0088, cluster 2 from 116 to 600 ms, p=0.0006; see *Figure 6—figure supplement 3* for complete cluster profiles). The results remained even when the transition-based surprise regressors were entered first, and then the regression on complexity was performed on the residuals (*Figure 6—figure supplement 2*, right column). In summary, the positive effect of complexity on habituation and standard trials, and its negative effect on deviant trials, were not solely due to local transition-based surprise signals.

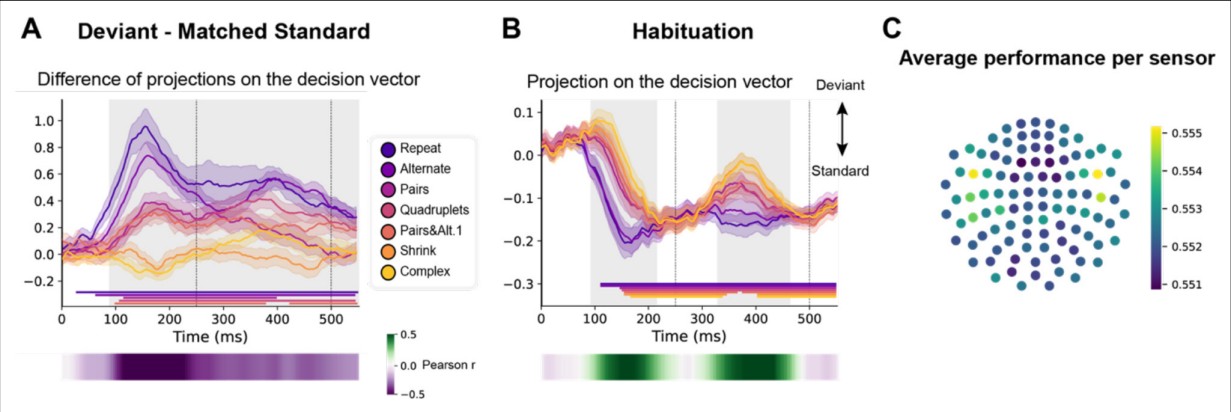

**Figure 7.** Multivariate decoding of deviant trials from magneto-encephalography (MEG) signals, and its variation with sequence complexity. (**A**) A decoder was trained to classify standard from deviant trials from MEG signals at a given time point. We here show the difference in the projection on the decision vector for *standard* and *deviant* trials, that is a measure of the decoder's accuracy. The decoder was trained jointly on all sequences, but its performance is plotted here for left-out trials separately for each sequence type. Shaded areas indicate s.e.m. and colored lines at bottom indicate the time windows identified by the temporal cluster-based permutation test (p<0.05 corrected) obtained from cluster-based permutation test on the full window. The heatmap at the bottom represents the correlation of the performance with sequence complexity (Pearson's r). The gray shaded time window in the main graph indicates the time window identified by the two-tailed p<0.05, temporal cluster-based permutation test. (**B**) Projection on the decision vector for *habituation* trials. The early brain response is classified as deviant but later as standard. This projection time course is increasingly delayed as a function of sequence complexity (same format as A). (**C**) Sensor map showing the relative contribution of each sensor to overall decoding performance. At the time of maximal overall decoding performance (165 ms) we trained and tested 4000 new decoders that used only a subset of 40 gradiometers at 20 sensor locations. For each sensor location, the color on the maps in the right column indicates the average decoding performance when this sensor location was used in decoding, thus assessing its contribution to overall decoding.

## Time-resolved decoding of violation responses

The above results were obtained by averaging sensor data across successive stimuli and across participants. A potentially more sensitive analysis method is multivariate decoding. It is a manner similar to *King and Dehaene, 2014*, which searches, at each time point and within each participant, for an optimal pattern of sensor activity reflecting a given type of mental representation. Therefore, to further characterize the brain representations of sequence structure and complexity, we next used multivariate time-resolved analyses, which allowed us to track sequence coding for each item in the sequence, at the millisecond scale.

We trained a decoder to classify all standard versus all deviant trials (*El Karoui et al., 2015*; *King et al., 2013*). As the two versions of the same sequence were presented on two separated runs (respectively starting with sound 'A' or 'B'), we trained and tested the decoder in a leave-one-run out manner, thus forcing it to identify non-stimulus-specific sequence violation responses. In addition, and most importantly, we selected standard trials that matched the deviants' ordinal position, which was specific to each sequence (see *Figure 1*, orange lines). *Figure 7* shows the average projection on the decision vector of the classifier's predictions on left-out data for the different sequences, when tested on both position-matched *deviants* versus *standards* (*Figure 7A*) and on *habituation* trials (*Figure 7B*). Significance was determined by temporal cluster-based permutation tests.

Decoding of deviants reached significance for all sequences except for the most complex ones (*Shrinking* and *Complex*). For the simplest *Repeat* and *Alternate* sequences, which could be learned solely based on transition probabilities, a sharp initial mismatch response was seen, peaking at ~150 ms. For all other sequences, the decoder exhibited a later, slower, lower-amplitude and sustained development of above-chance performance, suggesting that deviant items elicit decodable long-lasting brain signals. A temporal cluster-based permutation test on Pearson correlation with sequence complexity showed that the decoding of violations significantly correlated with complexity (temporal cluster from 90 to 580 ms).

The time courses of the decoder performance on habituation trials also revealed a clear hierarchy in the time it took for the brain to decide that a given tone was not a deviant (*Figure 7B*). The seven curves were ordered by predicted sequence complexity. Thus, the decoder's classification as standard, quantified as the projection on the decision vector, decreased significantly with sequence complexity

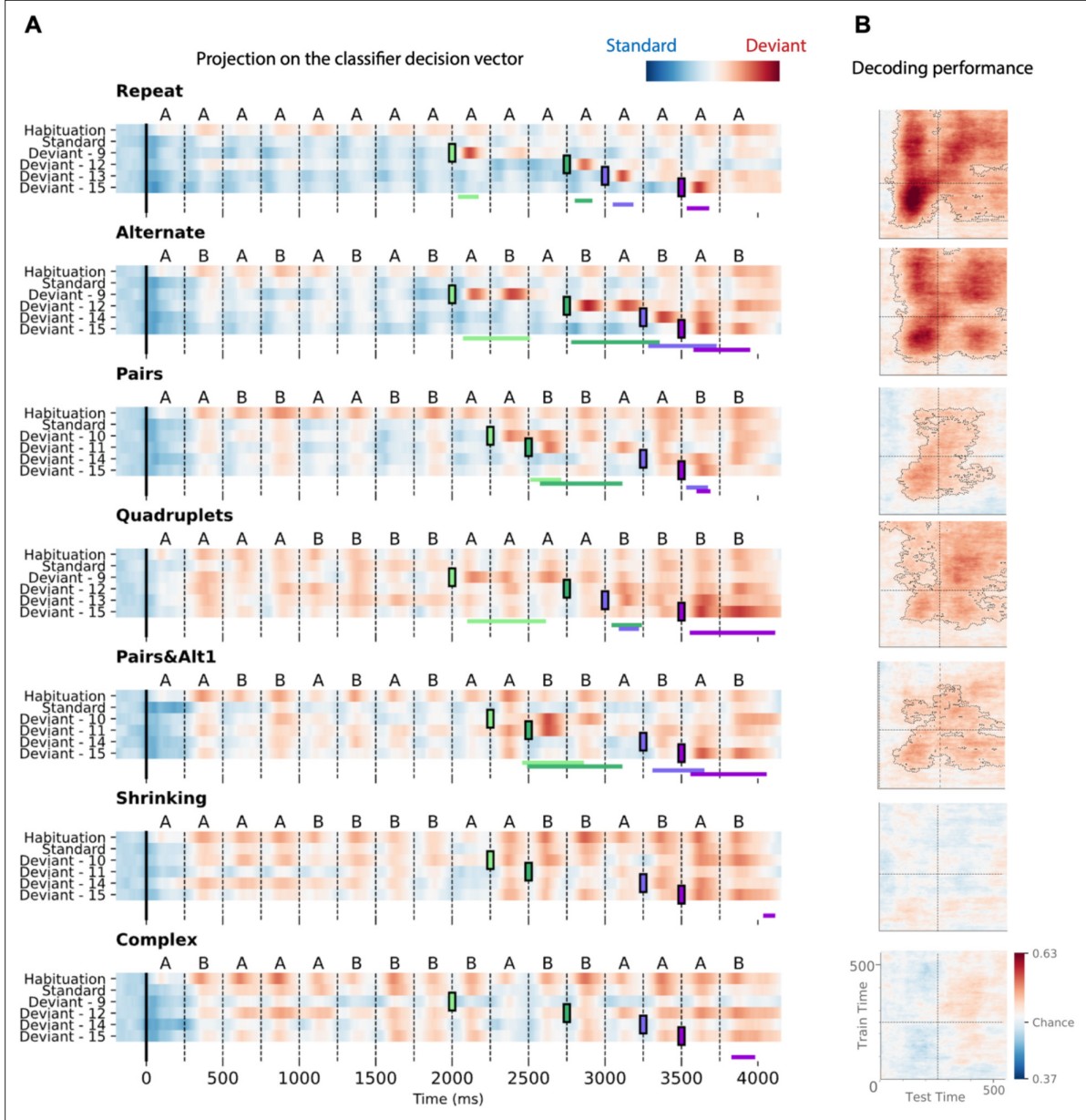

**Figure 8.** Time course of the deviancy decoder across the different types of sequences and deviant positions. (**A**) Average projection of magneto-encephalography (MEG) signals onto the decoding axis of the standard/deviant decoder. For each sequence, the time course of the projection was computed separately for habituation trials, standard trials, and for the four types of trials containing a deviant at a given position. The figure shows the average output of decoders trained between 130 and 210 ms post-deviant. Red indicates that a trial tends to be classified as a deviant, blue as a standard. Colored lines at the bottom of each graph indicate the time windows obtained from the cluster permutation test comparing deviants and standards in a 0–600 ms window after deviant onset. (**B**) Average generalization-across-time (GAT) matrices showing decoding performance as a function of decoder training time (y axis) and testing time (x axis). Vertical and horizontal lines indicate the onset of the next tone. The dashed lines outline p<0.05 cluster-level significance, corrected for multiple comparisons (see Materials and methods). Simpler sequences exhibit overall greater and more sustained performance. We note that, while deviancy detection does not reach significance for Shrinking and Complex sequences in the GAT matrices, violation signals reached significance for deviant position 15.

over two time windows (temporal cluster from ~90–220 ms and ~330–460 ms). This suggests that the more the sequence is complex, the more brittle its classification as standard is.

## Decoder performance over the full extent of each sequence

To characterize the time course of brain activity over the entire course of each sequence, we projected each MEG time point onto the decoding axis of the standard/deviant decoder trained on data from a 130–210 ms time window (*Figure 8*). The projection was computed separately for each sequence, separately for habituation, standard, and the four possible positions of deviant trials. We determined if deviants differed from standards using a cluster-based permutation test on a 0–600 ms window after each violation (colored lines at the bottom of each sequence in *Figure 8A*).

All individual deviants elicited a decodable response (see *Materials and methods*) except for the two highest-complexity sequences: *Shrinking* and *Complex* (failure at all positions exception the last one, i.e. 15). Interestingly, for the alternate sequence, two consecutive peaks indicate that, when a single repetition is introduced in an alternating sequence (e.g. ABABBBAB… instead of ABABABAB…), the brain interprets it as two consecutive violations, probably due to transition probabilities, as each of the B items is predicted to be followed by an A.

Most crucially, the analysis of specific violation responses allowed us to evaluate the range of properties that humans use to encode sequences, and to test the hypothesis that they integrate numerical and structural information at multiple nested levels (*Wang et al., 2015*). First, within a chunk of consecutive items, they detect violations consisting in both chunk shortening (one repeated tone instead of two in *Pairs*; three tones instead of four in *Quadruplets*) and chunk lengthening (three repeated tones instead of two, as well as five instead of four). The contrast between those two sequences clearly shows that participants possess a sophisticated context-dependent representation of each sequence. Thus, their brain emits a violation response upon hearing three consecutive items (AA**A**) within the *Pairs* sequence, where it is unexpected, but not when the same sequence occurs within the *Quadruplets* sequence. Conversely, participants are surprised to hear the transition BBAA**B** in the *Quadruplet* context, but not in the *Pairs* context. Finally, in the *Pairs+Alt.1* sequence, such context dependence changes over time, thus indicating an additional level of nesting: at positions 9–12, subjects expect to hear two pairs (AABB) and are surprised to hear A**B**BB (unexpected alternation), but just a second later, at positions 13–16, they expect an alternation (ABAB) and are surprised to hear A**A**AB (unexpected repetition). Similar, though less significant, evidence for syntax-based violation responses is present in the *Shrinking* sequence, which also ends with two pairs and an alternation.

*Figure 8* also shows in detail how the participants' brain fluctuates between predictability (in blue) and violation detection (in red) during all phases of the experiment. Initially, during habituation (top line), sequences are partially unpredictable, as shown by red responses to successive stimuli, but that effect is strongly modulated by complexity, as previously reported (red responses, particularly for the most complex sequences). In a sense, while the sequence is being learned, all items in those sequences appear as deviants. As expected, after habituation, the deviancy response to standards is much reduced, but still ordered by complexity. Higher-complexity sequences such as *Shrinking* thus create a globally less predictable environment (red colors) relative to which the violation responses to deviants appear to be reduced.

*Figure 8B* also shows how the standard-deviant decoder generalizes over time, separately for each sequence. The performance for the *Repeat* sequence exhibits a peak corresponding to the deviant item's presentation (~150 ms) and a large and a partial square pattern, indicating a sustained maintenance of the deviance information. The performance for the *Alternate* sequence shows four peaks spaced by the SOA, corresponding to the two deviant transitions elicited by the deviant item. *Pairs*, *Quadruplets,* and *Pairs+Alt.1* sequences still show significant decoding but not *Shrinking* and *Complex* sequences, indicating that the ability to decode deviant signals decreases with complexity.

## Discussion

The goal of this study was to characterize the mental representation that humans utilize to encode binary sequences of sounds in memory and to detect occasional deviants. The results indicate that, in the human brain, deviant responses go way beyond the sole detection of violations in habitual sounds (*May and Tiitinen, 2010*) or in transition probabilities (*Wacongne et al., 2012*), and are also

sensitive to more complex, larger-scale regularities (*Bekinschtein et al., 2009*; *Bendixen et al., 2007*; *Maheu et al., 2019*; *Schröger et al., 2007*; *Wacongne et al., 2011*; *Wang et al., 2015*). Instead of merely storing each successive sound in a distinct memory slot (*Baddeley, 2003*; *Baddeley and Hitch, 1974*; *Botvinick and Watanabe, 2007*; *Hurlstone et al., 2014*), behavioral and brain imaging results suggest that participants mentally compressed these sequences using an algorithmic-like description where sequence regularities are expressed in terms of recursive combinations of simple rules (*Al Roumi et al., 2021*; *Dehaene et al., 2015*; *Planton et al., 2021*). Consistently with the predictions of this formal LoT, behavioral performance and brain responses were modulated by the MDL of the sequence, which we term LoT complexity. We discuss those points in turn.

Behavioral results during fMRI fully replicated our previously behavioral work (*Planton et al., 2021*). First, performance in detecting occasional deviants, thus indexing sequence memory, was strongly modulated by MDL in our formal language (LoT complexity; *Figure 2A*, top). Second, even when a deviant was correctly detected, response time was strongly correlated with LoT complexity (*Figure 2A*, bottom). Both findings indicate that novelty detection mechanisms were impacted by sequence structure. Finally, after the experiment, when participants were asked to segment the sequences with brackets, their segmentations closely matched the LoT sequence descriptions. For instance, they segmented the *Pairs+Alt.1* sequence as [[AA][BB]][[ABAB]]. The sole exception was the alternate sequence (ABAB...) which was encoded by our theory as 15 alternations, but was bracketed by participants as 8 repetitions of the subsequence AB. This interesting departure from our theory may indicate that, during sequence parsing, participants do not necessarily identify the most compact representation, but wait until a repeated subsequence occurs, and then encode how often it repeats (including nested repetitions). This parsing strategy would yield a minor departure from our proposed encodings. Undoubtedly, there is still room for improvement in our LoT theory, which is highly idealized and does not incorporate real-time parsing constraints. However, an alternative interpretation of the bracketing results is that the visual bracketing task itself may bias the perception of auditory sequences. By making the entire sequence visible at once, including its start and end point, the visual format may have incited subjects to subdivide it into groups of two, while the auditory sequential presentation alone would have encouraged the '15 alternations' encoding. In support of the latter interpretation, we did not find any evidence for grouping by two in the timing of button presses when subjects reproduced the alternate sequence from memory (Al Roumi and Tabbane, unpublished data). More research, with a greater diversity of sequences and tasks, will be needed to understand whether and where the present theory needs to be amended.

Altogether, these behavioral results confirm that the postulated LoT provides a plausible description of how binary sequences are encoded. They fit with a long line of cognitive psychological research searching for computer-like languages that capture the human concept of sequence regularity (*Leeuwenberg, 1969*; *Restle, 1970*; *Restle and Brown, 1970*; *Simon, 1972*; *Simon and Kotovsky, 1963*). Here, as in *Planton and Dehaene, 2021*, a formal statistical comparison demonstrated the superiority of LoT complexity against many competing measures such as transition probability, chunk complexity, entropy, subsymmetries, Lempel-Ziv compression, change complexity, or algorithmic complexity. In the next sections, we discuss how brain imaging results provide additional information on how sequence compression is implemented in the human brain.

According to our hypothesis, the more complex the sequence, the longer the internal model and the larger the effort to parse it, encode it and maintain it in working memory. Consequently, we expected during the habituation phase larger brain activations for more complex sequences in regions that are involved in auditory sequence encoding. Both fMRI and MEG results supported this hypothesis. Importantly, contrary to the fMRI experiment, the MEG experiment did not require overt responses, yet several neural markers, such as global field power, showed a significant increase with sequence complexity (*Figure 6A*). Furthermore, linear regressions showed that brain activity increased with sequence complexity for sensors that corresponded to the auditory and inferior frontal regions (*Figure 6B*).

Many levels of sequence-processing mechanisms coexist in the human brain (*Dehaene et al., 2015*) and statistical learning is a well-known contributor to brain activity. Thanks to the high temporal resolution of MEG, we could separate the effects of transition probabilities from those of sequence structure (*Bekinschtein et al., 2009*; *Maheu et al., 2019*; *Wacongne et al., 2011*). To separate them, we ran a multilinear regression model with regressors for both. Even after adding four additional regressions

for immediate and longer-term transition statistics, a cluster-based permutation test provided similar spatiotemporal clusters for the regressor for LoT complexity (*Figure 6—figure supplement 2* and *Figure 6—figure supplement 3*). As shown in *Figure 6—figure supplement 4*, for habituation and standard trials, repetition/alternation impacted on both an early peak at 80 ms and a later one at 170 ms after stim onset, perhaps reflecting sensory bottom-up versus top-down processes. Transition-based surprise exhibited only one peak at ~110 ms after stim onset. The 20 ms delay between the peaks supports the possibility that the first reflects low-level neural adaptation while the second corresponds to a violation of expectations based on transition probabilities. Complexity effects, however, showed a later and more sustained response, extending much beyond 200 ms for deviant trials, in agreement with a distinct rule-based process.

Previous fMRI results led us to expect several prefrontal regions to exhibit an increasing activity with sequence complexity (*Badre, 2008*; *Badre et al., 2010*; *Barascud et al., 2016*; *Koechlin et al., 2003*; *Koechlin and Jubault, 2006*; *Wang et al., 2019*), but no such activation was observed in MEG source reconstruction. This negative result has several potential explanations. First of all, sequence complexity may act as a context effect and therefore may be sustained across time (*Barascud et al., 2016*; *Southwell and Chait, 2018*). As we baselined the data on a short time window before each sound onset, such a constant effect may be removed. Furthermore, frontal brain regions may be too distributed, intermixed, and/or too far from the MEG helmet to be faithfully reconstructed. Finally, the fMRI experiment allowed us to clearly identify a large network of brain areas involved in complexity, but recruiting a rather posterior region of prefrontal cortex, the preCG (or dorsal premotor cortex, PMd, bordering on the dorsal part of Brodmann area 44) together with the STG, SMA, cerebellum, and IPS that all exhibited the predicted increase in activity with LoT complexity. All these regions showed the predicted increasing response with complexity during habituation, and decreasing response with complexity to deviants.

All these areas have been shown to be associated with temporal sequence processing, although mostly with oddball paradigms using much shorter or simpler sequences (*Bekinschtein et al., 2009*; *Huettel et al., 2002*; *Planton and Dehaene, 2021*; *Wang et al., 2015*; *Wang et al., 2019*). They can be decomposed into modality-specific and modality-independent regions (*Frost et al., 2015*). STG activation was observed for auditory sequences here and in other studies (*Bekinschtein et al., 2009*; *Wang et al., 2015*) but not for visuo-spatial sequences (*Wang et al., 2019*). The modality specificity of STG was explicitly confirmed by *Planton and Dehaene, 2021*, using visual and auditory sequences with identical structures. Other regions, meanwhile, were modality-independent and coincided with those found in a similar paradigm with visuo-spatial sequences (*Wang et al., 2019*), consistent with a role in abstract rule formation. The IPS and preCG, in particular, are jointly activated in various conditions of mental calculation and mathematics (*Amalric et al., 2017b*; *Dehaene et al., 2003*), with anterior IPS housing a modality-invariant representation of number (*Dehaene et al., 2003*; *Eger et al., 2009*; *Harvey et al., 2013*; *Kanayet et al., 2018*). The overlap between auditory sequences and arithmetic was confirmed here using sensitive single-subject analyses (*Figure 5*). PreCG and IPS may thus be jointly involved in the nested 'for i=1:n' loops, i.e. the repetitions of the proposed language, and in the real-time tracking of item and chunk number needed to follow a given auditory sequence even after it was learned. Such a role is consistent with the more general proposal of a role for these dorsal regions, particularly the IPS, in structure learning (*Summerfield et al., 2020*). While these regions coactivated with STG during the present auditory task, in a previous visuo-spatial version of the same task they did so together with bilateral occipito-parietal areas (*Wang et al., 2019*). This is consistent with the behavioral observation that the very same language, involving concatenation, loops, and recursion, when applied to visual or auditory primitives, can account for sequence memory in both modalities (*Dehaene et al., 2022*; *Planton et al., 2021*).

Our data also point to the SMA, or rather pre-SMA (*Nachev et al., 2008*), in processing increasingly complex sequences. Such a domain-general sequence-processing function was indeed advocated by *Cona and Semenza, 2017*, given its various involvements in action sequences, music processing, numerical cognition, spatial processing, time processing, as well as language. Remarkably, the cerebellum also participated in our complexity network. Its role in working memory has been rarely reported or discussed and might have been underestimated in the parsing of non-motor sequences, as it is classically associated with motor sequence learning (*Jenkins et al., 1994*; *Toni et al., 1998*). The present results confirm that the cerebellum may be involved in abstract, non-motor sequence encoding and

expectation (*Leggio et al., 2008*; *Molinari et al., 2008*; *Nixon, 2003*). Indeed, cerebellum, SMA, and premotor cortex were already reported as involved in the passive listening of rhythms (*Chen et al., 2008*), consistent with a role in the identification of sequence regularities. A tentative hypothesis is that (pre)SMA, cerebellum, and possibly premotor cortex may participate in a beat- (*Morillon and Baillet, 2017*) or time-processing network (*Coull et al., 2011*), thus possibly involved in the translation from the abstract structures of the proposed language to concrete, precisely timed sensory predictions.

Interestingly, we found that, while task performance was primarily linearly related to LoT complexity, fMRI activity was not. Rather, as the sequence becomes too complex, activation tended to stop increasing, or even decreased, just yielding a significant downward quadratic trend. *Wang et al., 2019*, observed a similar effect with visuo-spatial sequences. In both cases, the highest complexity sequences were largely incompressible because they did not have any significant regularity in our language and, given their length, could not be easily memorized. The collapse of activity at a certain level of LoT complexity, in regions that are precisely involved in working memory, is therefore logical. Indeed, in a more classical object memory task, *Vogel and Machizawa, 2004*, found that working memory activity does not solely increase with the number of elements stored in working memory, but saturates or decreases once the stimuli exceed storage capacity, thought to be around three or four items (*Cowan, 2001*; for discussion, see *Ma et al., 2014*). Naturally, such a collapse can only lead to reduced predictions and therefore reduced violation detection – thus explaining that behavioral responses to deviants continue to increase linearly with complexity, while model-related fMRI activations vary as an inverted U function of complexity. An analogous phenomenon was described in infants (*Kidd et al., 2012*; *Kidd et al., 2014*): they allocate their attention to visually or auditory presented sequences that are neither too simple nor too complex, thus showing a U-shaped pattern that implies boredom for stimuli with low information content and saturation from stimuli that exceed their cognitive resources.

Detailed examination of the responses to violations in MEG confirmed that human participants were able to encode details of the hierarchical structures of sequences. Not only did the amplitude of violation responses tightly track the proposed LoT complexity (*Figure 7*), but the specific violation responses proved that the human brain changed its expectations in a hierarchical manner (*Figure 8*). This was clearest in the case of the *Pairs+Alt1* sequence, which consists in two pairs (AABB) followed by four alternations (ABAB). In those two consecutive parts, the predictions are exactly opposite at central locations (A**AB**B versus A**BA**B), such that what is a violation for one is a correct prediction for the other, and vice versa. The fact that we observe significant violation responses at each of these locations (i.e. locations 10, 12, 14, and 15 in the *Pairs+Alt1* sequence), as well as for the matched *Alternate* and *Pairs* sequences, indicates that the human brain is able to quickly change its anticipations as a function of sequence hierarchical structure. To do so, it must contain a representation of sequences as nested parts within parts, and switch between those parts after a fixed number of items (4 in this case). Violation detection in the *Pairs* and *Quadruplets* sequences further confirmed that subjects kept track of the exact number of items in each subsequence, since their brain reacted to violations that either shortened or lengthened a chunk of identical consecutive items.

While present and past results indicate that a language is necessary to account for the human encoding of binary auditory sequences (*Dehaene et al., 2022*; *Planton et al., 2021*), this language differs from those used for communication: it involves repetitions, numbers, and symmetries, while the syntax of natural language systematically avoids these features (*Moro, 1997*; *Musso et al., 2003*). In agreement with this observation, there was little overlap between our auditory sequence complexity network and the classical left-hemisphere language network. Instead, complexity effects were systematically distributed symmetrically in both hemispheres, unlike natural language processing. Within individually defined language functional ROIs (fROIs) (defined by their activity during visual or auditory sentence processing relative to a low-level control), no significant complexity effect was found except in a single region, the left IFGoper (a negative effect of complexity for deviants was also found there and in pSTS). Even that finding may well be a partial volume effect, as this area was absent from whole-brain contrasts, and the centroid of the complexity-related activation was centered at a more dorsal location in preCG (*Figure 3*). Broca's area is the main candidate region for language-like processing of hierarchical structures, and such role is advocated for in various previous rule-learning studies using artificial grammars (*Bahlmann et al., 2008*; *Fitch and Friederici, 2012*; *Friederici*

*et al., 2006*), structured sequences of actions (*Badre and D'Esposito, 2007*; *Koechlin and Jubault, 2006*), sequence processing (*Wang et al., 2015*), and even music (*Maess et al., 2001*; *Patel, 2003*). However, Broca's area is a heterogeneous region (*Amunts et al., 2010*), of which certain sub-regions support language while others underlie a variety of other cognitive functions, including mathematics and working memory (*Fedorenko et al., 2012*). Interpretation must remain careful since functions that were once thought to overlap in Broca's area, such as language and musical syntax (*Fadiga et al., 2009*; *Koelsch et al., 2002*; *Kunert et al., 2015*), are now clearly dissociated by higher-resolution single-subject analyses (*Chen et al., 2021*).

Conversely, a very different picture was observed when examining the overlap of LoT complexity fMRI activity and the mathematical calculation network. There was considerable overlap at the whole-brain level (SMA, IPS, premotor cortex, cerebellum) and, most importantly, a significant sequence complexity effect within each of the individual mathematical fROIs. A similar result was reported by *Wang et al., 2019*; they found activation of mathematics-related regions but not language-related ones when participants were processing visuo-spatial sequences. *Planton and Dehaene, 2021*, reached a similar conclusion by showing novelty effects to pattern violations of both visual and auditory short sequences in mathematics but not in language areas. Since their data, as well as the present data, was obtained with binary sequences which, contrary to *Wang et al., 2019*, were devoid of any geometrical content, those results indicate that the amodal language of thought for sequences shares common neural mechanisms with mathematical thinking, even when no overt geometrical content is present.

The present results therefore support the hypothesis that the human brain hosts multiple internal languages, depending on the types of structures and contents that are being processed (*Dehaene et al., 2022*; *Fedorenko and Varley, 2016*; *Hagoort, 2013*). While the capacity to encode nested sequences may well be a fundamental overarching function of the human brain, fundamental to the manipulation of hierarchical structures in language, mathematics, music, complex actions, etc. (*Dehaene et al., 2015*; *Fitch, 2014*; *Hauser et al., 2002*; *Lashley, 1951*), those abilities may rely on partially dissociable networks. This conclusion fits with much prior evidence that, at the individual level, language and mathematics do not share the same cerebral substrates and may be dissociated by brain injuries (*Amalric and Dehaene, 2016*; *Fedorenko and Varley, 2016*), just like language and music (*Chen et al., 2008*; *Norman-Haignere et al., 2015*; *Peretz et al., 2015*). During hominization, we speculate that an enhanced functionality for recursive nesting may have jointly emerged in all of those neuronal circuits (*Dehaene et al., 2022*). In the future, this hypothesis could be tested by submitting non-human primates to the present hierarchy of sequences, and examine up to which level their brains can react to violations. We already know that the macaque monkey brain can detect violations of simple habitual, sequential, or numerical patterns (*Uhrig et al., 2014*; *Wilson et al., 2013*), with both convergence (*Wilson et al., 2017*) and divergence (*Wang et al., 2015*) relative to human results. The present design may help determine precisely where to draw the line.

## Materials and methods

### Participants
Nineteen participants (10 men, $M_{age}$ = 27.6 years, $SD_{age}$ = 4.7 years) took part in the MEG experiment and 23 (11 men, $M_{age}$ = 26.1 years, $SD_{age}$ = 4.7 years) in the fMRI experiment. The two groups of participants were distinct. We did not test any effect of gender on the results of this study. All participants had normal or corrected to normal vision and no history or indications of psychological or neurological disorders. In compliance with institutional guidelines, all subjects gave written informed consent prior to enrollment and received 90€ as compensation. The experiments were approved by the national ethical committees (CPP Ile-de-France III and CPP Sud-Est VI).

### Stimuli and tasks
Auditory binary sequences of 16 sounds were used in both experiments. They were composed of low-pitch and high-pitch sounds, constructed as the superimposition of sinusoidal signals of respectively $f$=350 Hz, 700 Hz, and 1400 Hz, and $f$=500 Hz, 1000 Hz, and 2000 Hz. Each tone lasted 50 ms and the 16 tones were presented in sequence with a fixed SOA of 250 ms.

Ten 16-item sequential patterns spanning a large range of complexities were selected (see *Figure 1A*). Six of them were used in previous behavioral experiments (*Planton et al., 2021*). The complexity metric used to predict behavior and brain activity was the 'LoT – *chunk*' complexity, which was previously shown to be well correlated with behavior (*Planton et al., 2021*). This metric roughly measures the length of the shortest description of the pattern in a formal language that uses a small set of atomic rules (e.g. repetition, alternation) that can be recursively embedded. The *chunk* version of the metric includes only expressions that preserve chunks of consecutive repeated items (for instance, the sequence ABBA is parsed as [A][BB][A] rather than [AB][BA]). 10 sequences were used in the fMRI experiment, and 7 of them in the MEG experiment (i.e. all but *Pairs&Alt.2*, *ThreeTwo*, and *CenterMirror*).

Each auditory sequence (4000 ms long) was repeatedly presented to a participant in a mini-session with 500 ms ITI. Mini-sessions had the following structure. Participants first discovered and encoded the sequence during a habituation phase of 10 trials. Then, during a test phase, occasional violations consisting in the replacement of a high-pitch sound by a low-pitch one (or vice versa) were presented at the locations specified in *Figure 1A*. As described in *Figure 1B*, in the MEG experiment, the test phase included 36 trials of which 2/3 comprised a deviant sound. In the fMRI experiment, the test phase included 18 trials of which 1/3 comprised a deviant sound. Participants were unaware of the mini-session structure.

In the MEG experiment, habituation and test sequences followed each other seamlessly, and participants were merely asked to listen attentively. After each mini-session, they were asked one general question about what they had just heard such as: *How many different sounds could you hear? Did you find it musical? How complex was the sequence of sounds?* The full experiment was divided temporally into two parts such that the seven sequence types appeared twice, once in each version (starting with A or B), once at the beginning and once at the end of the experiment. The overall experiment lasted about 80 min.

In the fMRI experiment, participants were explicitly instructed to detect and respond to violations, by pressing a button, as quickly as possible, with either their right or left hand. The correct response button (left or right, counterbalanced over the two repetitions of each sequence) was indicated by a 2 s visual message on the screen during the rest period preceding the first test trial. In order to optimize the estimation of the BOLD response, trials were presented in two blocks of five trials for the habituation phase, then three blocks of six trials for the test phase, separated by rest periods of variable duration (6 s±1.5). The 10 sequences appeared twice, once in each version (starting with A or B). The 20 mini-sessions were presented across five fMRI sessions of approximately 11 min.

## Post-experimental sequence bracketing task

After the experiment, participants were given a questionnaire to assess their own representation of the structure of the sequence. For each sequence of the experiment (i.e. 7 for the MEG participants, 10 for the fMRI participants), after listening to it several times if needed, participants were asked to segment the sequence by drawing brackets (opening and closing) on its visual representation (As and Bs were respectively represented by empty and filled circles on a sheet of paper). In this way, they were instructed to indicate how they tended to group consecutive items together in their mind when listening to the sequence, if they did.

## fMRI experiment procedures

### Localizer session

Together with the main sequence-processing task described above, the fMRI experimental protocol also included a 6 min localizer session designed to localize cerebral regions involved in language processing and in mathematics. This localizer was already used in our previous work (*Planton and Dehaene, 2021*) and is a variant of a previously published functional localizer which is fully described elsewhere (*Pinel et al., 2007*). A sentence-processing network was identified in each subject by contrasting sentence reading/listening conditions (i.e. visually and auditorily presented sentences) from control conditions (i.e. meaningless auditory stimuli consisting in rotated sentences, and meaningless visual stimuli of the same size and visual complexity as visual words). A mathematics network was identified in each subject by contrasting mental calculation conditions (i.e. mental processing of

simple subtraction problems, such as 7–2, presented visually, and auditorily) from sentence reading/listening conditions.

## fMRI acquisition and preprocessing

MRI acquisition was performed on a 3T scanner (Siemens, Tim Trio), equipped with a 64-channel head coil. 354 functional scans covering the whole brain were acquired for each of the five sessions of the main experiment, as well as 175 functional scans for the localizer session, all using a T2*-weighted gradient echo-planar imaging sequence (69 interleaved slices, TR = 1.81 s, TE = 30.4 ms, voxel size = 1.75 mm$^3$, multiband factor = 3). To estimate distortions, two volumes with opposite phase encoding direction were acquired: one volume in the anterior-to-posterior direction (AP) and one volume in the other direction (PA). A 3D T1-weighted structural image was also acquired (TR = 2.30 s, TE = 2.98 ms, voxel size = 1.0 mm$^3$).

Data processing (except the TOPUP correction) was performed with SPM12 (Wellcome Department of Cognitive Neurology, http://www.fil.ion.ucl.ac.uk/spm). The anatomical scan was spatially normalized to a standard Montreal Neurological Institute (MNI) reference anatomical template brain using the default parameters. Functional images were unwarped (using the AP/PA volumes, processed with the TOPUP software; FSL, fMRIB), corrected for slice timing differences (first slice as reference), realigned (registered to the mean using second-degree B-splines), coregistered to the anatomy (using normalized mutual information), spatially normalized to the MNI brain space (using the parameters obtained from the normalization of the anatomy), and smoothed with an isotropic Gaussian filter of 5 mm FWHM.

In addition to the 6 motion regressors from the realignment step, 12 regressors were computed using the aCompCor method (*Behzadi et al., 2007*), applied to the CSF and to white matter (first five components of two principal component analyses, and one for the raw signal), in order to better correct for motion-related and physiological noise in the statistical models (using the PhysIO Toolbox; *Kasper et al., 2017*). Additional regressors for motion outliers were also computed (framewise displacement larger than 0.5 mm; see *Power et al., 2012*), they represented 0.5% of volumes per subject on average. One participant was excluded from the fMRI analyses due to excessive movement in the scanner (average translational displacement of 2.9 mm within each fMRI session, which was 3.3 SD above group average).

## fMRI analysis
### General linear model

Statistical analyses were performed using SPM12 and GLM that included the motion-related and physiological noise-related regressors (described above) as covariates of no interest. fMRI images were high-pass filtered at 0.01 Hz. Time series from the sequences of stimuli of each condition (each tone modeled as an event) were convolved with the canonical hemodynamic response function. Specifically, for each of the 20 mini-sessions (i.e. each sequence being tested twice, reverting the attribution of the two tones), one regressor for the items of the habituation phase, one for the items of the test phase, and one for the deviant items were included in the GLM. Since motor responses and deviant trials were highly collinear, manual motor responses were not modeled. However, motor responses could be less frequent for more complex sequences (i.e. increased miss rate), thus creating a potential confound with the effect of complexity in deviant trials. We thus also computed an alternative model in which only correctly detected deviants trials were included. In order to test for a relationship between brain activation and LoT complexity in different trial types (i.e. habituation trials, deviant trials), corresponding beta maps for each of the 10 sequences and each participant were entered in second-level within-subject ANOVA. Linear parametric contrasts using the LoT complexity value were then computed.

### Cross-validated ROI plots and analyses

To further test the reliability of the complexity effect across participants, a cross-validated ROI analysis, using individually defined fROIs, was conducted. Nine of the most salient peaks from the positive LoT complexity contrast in habituation were first selected, and used to build nine 20-mm-diameter spherical search volumes: SMA (coordinates: –1, 5, 65), right precentral gyrus (R-preCG; 46, 2, 44), left precentral gyrus (L-preCG; –47, 0, 45), right intraparietal sulcus (R-IPS;

36, −46, 56), left intraparietal sulcus (L-IPS; −31, −42, 44), right superior temporal gyrus (R-STG; 48, −32, 3), left superior temporal gyrus (L-STG; −68, −23, 5), lobule VI of the left cerebellar hemisphere (L-CER6; −29, −56, −28) and lobule VI of the right cerebellar hemisphere (R-CER8; 22, −68, −51). Since these search volumes were defined on group-level results, to reduce the degree of circularity when extracting the individual ROI data, these were extracted in a cross-validated manner, by separating each participant data into two halves. Individual fROIs were then defined for each participant by selecting the 20% most active voxels at the intersection between each search volume and the contrast 'LoT complexity effect in habituation' computed on half of the blocks (i.e. blocks of sequences starting with 'A'). Mean contrast estimates for each fROI and each condition was then extracted using the other half of the blocks (i.e. blocks of sequences starting with 'B'). The same procedure was repeated a second time by reversing the role of the two halves (i.e. fROIs computed using blocks of sequences starting with 'B', data extracted from blocks of sequences starting with 'A'). It should be noted that this procedure does not fully eliminate the circularity in the analysis, since the initial search volumes were still based on whole-group data – however, such circularity should be minimal, as the ROIs themselves. To test for the significance of the complexity effect in each ROI, the mean of the output of the two procedures (i.e. the cross-validated activation value), for each of the 10 conditions (i.e. habituation blocks for each of the 10 sequences) and each participant, was entered in a linear mixed-effect model with participant as random factor and LoT complexity value as a fixed-effect predictor. p-Values were corrected for multiple comparison using Bonferroni correction for nine ROIs. Along with such a linear effect of complexity, we also tested a quadratic effect by adding a quadratic term in the mixed-effect model.

In order to track activation over time, we also extracted, using the same cross-validated procedure, the BOLD activation time course for each 28-trial mini-session. To account for the fact that the duration of rest periods between blocks could vary, data were actually extracted for a [−6 s −32 s] period relative to the onset of the first trial of each block rest period, and the whole mini-session curve was recomposed by averaging over the overlapping period of two consecutive parts (see vertical shadings in *Figure 3B*). Each individual time course was upsampled and smoothed using cubic spline interpolation, and baseline corrected with a 6 s period preceding the onset of the first trial.

Finally, two set of ROIs were selected in order to test for the involvement of language and mathematics-related areas in the present sequence-processing task, and especially to assess a potential sequence complexity effect. Seven language-related ROIs came from the sentence-processing experiment of *Pallier et al., 2011*: pars orbitalis (IFGorb), triangularis (IFGtri), and opercularis (IFGoper) of the inferior frontal gyrus, TP, TPJ, aSTS, and pSTS. Seven mathematics-related ROIs came from the mathematical thinking experiment of *Amalric and Dehaene, 2016*: left and right IPS, left and right SFG, left and right precentral/inferior frontal gyrus (preCG/IFG), SMA. These two sets of ROIs were already used in the past (*Planton and Dehaene, 2021*; *Wang et al., 2019*). In order to build individual and functional ROIs from these literature-based ROIs, we used the same procedure as *Planton and Dehaene, 2021*, consisting in selecting, for each subject, the 20% most active voxels within the intersection between the ROI mask and an fMRI contrast of interest from the independent localizer session. The contrast of interest was 'Listening & reading sentences > Rotated speech & false font script' for the ROIs of the language network, and 'Mental calculation visual & auditory > Sentence listening & reading' for the ROIs of the mathematics network. Mean contrast estimates for each fROI and each condition was then extracted, and entered into linear mixed-effect model with participant as random factor and LoT complexity value as a fixed effect predictor. A Bonferroni correction for 14 ROIs was applied to the p-values.

## Behavioral data analysis

Data for the sequence bracketing task included all productions collected in the fMRI and MEG experiment (42 participants for seven sequences, and 23 participants for the three that were only presented during fMRI). For each production, we counted the total number of brackets (opening and closing) drawn at each interval between two consecutive items (as well as before the first and after the last item, resulting in a vector of length 17) (see *Figure 2A*). To determine if participants' reported sequence structure matched the predictions of the LoT model, we computed the correlation between the average over participants of the number of brackets in each interval and the postulated bracketing of the sequence (derived from its expression in the LoT). For the

first two sequences, the representations '[A][A][A]…' and '[AAA…]', as well as '[A][B][A]…', and '[ABA…]', respectively, derived from the expressions [+0]^16 and [+0]^16<b >, were considered as equivalent.

For the violation detection task of the fMRI experiment, we considered as a correct response (or 'hit') all button presses occurring between 200 ms and 2500 ms after the onset of a deviant sound. We thus allowed for potential delayed responses (but found that 97.7% of correct responses were below 1500 ms). An absence of response in this interval was counted as a miss, a button press outside this interval was counted as a false alarm. We then computed, for each subject and each sequence, the average response time as well as, using the proportions of hits and false alarms, the sensitivity (or *d'*). The method of *Hautus, 1995*, was used to adjust extreme values. To test whether subject performance correlated with LoT complexity, we performed linear regressions on group-averaged data, as well as linear mixed models including participant as the (only) random factor. The random effect structure of the mixed models was kept minimal, and did not include any random slopes, to avoid the convergence issues often encountered when attempting to fit more complex models. Analyses were performed in R 4.0.2 (*R Development Core Team, 2020*), using the lme4 (*Bates et al., 2015*) and lmerTest (*Kuznetsova et al., 2017*) packages. Surprise for each deviant item was computed from transition probabilities, within each block for each subject, using an ideal observer Bayesian model (*Maheu et al., 2019*; *Meyniel et al., 2016*) and tested as an additional predictor in the mixed-effect models. For the analysis of *d'*, we used the average surprise of the deviant items of the block (i.e. all deviants presented to the subject, whether or not they detected them). For the analysis of response times, we used the average transition-based surprise of the correctly detected deviant items of the block.

## MEG experiment procedures

### MEG recordings

Participants listened to the sequences while sitting inside an electromagnetically shielded room. The magnetic component of their brain activity was recorded with a 306-channel, whole-head MEG by Elekta Neuromag (Helsinki, Finland). 102 triplets, each comprising one magnetometer and two orthogonal planar gradiometers composed the MEG helmet. The brain signals were acquired at a sampling rate of 1000 Hz with a hardware high-pass filter at 0.1 Hz. The data was then resampled at 250 Hz.

Eye movements and heartbeats were monitored with vertical and horizontal electro-oculograms (EOGs) and electrocardiograms (ECGs). Head shape was digitized using various points on the scalp as well as the nasion, left and right pre-auricular points (FASTTRACK, Polhemus). Subjects' head position inside the helmet was measured at the beginning of each run with an isotrack Polhemus Inc system from the location of four coils placed over frontal and mastoïdian skull areas. Sounds were presented using Eatymotic audio system (an HiFi-quality artifact-free headphone system with wide-frequency response) while participants had to fixate a central cross. The analysis was performed with MNE Python (*Gramfort et al., 2013*; *Jas et al., 2018*), version 0.23.0.

### Data cleaning: maxfiltering

We applied the signal space separation algorithm *mne.preprocessing.maxwell_filter* (*Taulu et al., 2004*) to suppress magnetic signals from outside the sensor helmet and interpolate bad channels that we identified visually in the raw signal and in the power spectrum. This algorithm also compensated for head movements between experimental blocks by realigning all data to an average head position.

### Data cleaning: independent component analysis

Oculomotor and cardiac artifacts were removed performing an independent component analysis on the four last runs of the experiment. The components that correlated the most with the EOG and ECG signals were automatically detected. We then visually inspected their topography and correlation to the ECG and EOG time series to confirm their rejection from the MEG data. A maximum of one component for the cardiac artifact and two components for the ocular artifacts were considered. Finally, we removed them from the whole recording (14 runs).

## Data cleaning: autoreject

We used an automated algorithm for rejection and repair of bad trials (*Jas et al., 2017*) that computes the optimal peak-to-peak threshold per channel type in a cross-validated manner. It was applied to baselined epochs and removed on average 4.6% of the epochs.

## Epoching parameters and projection on magnetometers

Epochs on items were baselined from –50 to 0 ms (stimulus onset) and epochs on the full sequences were baselined between –200 and 0 ms (first sequence item onset). For sensor level analyses, instead of working with the 306 sensors (102 magnetometers and 206 gradiometers), we projected the spherical sources of signal onto the magnetometers using MNE epochs method *epochs.as_type('mag',mode='accurate')*.

## Univariate analyses

### GFP and linear regressions

Global field power was computed as the root-mean-square of evoked responses or the difference of evoked responses. Linear regressions were computed using fourfold cross-validation and with the *linear_model.LinearRegression* function of scikit-learn package version 0.24.1. Pearson correlation was computed with the *stats.pearsonr* function from *scipy* package. The predictors for surprise from transition probabilities were computed using an ideal observer Bayesian model learning first-order transitions with an exponential memory decay over 100 items. This was done thanks to the Transition-ProbModel python package, which is the python version of the *Matlab version* used in *Maheu et al., 2019*; *Meyniel et al., 2016*.

### Source reconstruction

A T1-weighted anatomical MRI image with 1 mm isometric resolution was acquired for each participant (3T Prisma Siemens scanner). The anatomical MRI was segmented with FreeSurfer (*Dale et al., 1999*; *Fischl et al., 2002*) and co-registered with MEG data in MNE using the digitized markers. A three-layer boundary element model (inner skull, outer skull, and outer skin) was used to estimate the current-source density distribution over the cortical surface. Source reconstruction was performed on the linear regression coefficients using the dSPM solution with MNE default values (loose orientation of 0.2, depth weighting of 0.8, SNR value of 3) (*Dale et al., 2000*). The noise covariance matrix used for data whitening was estimated from the signal within the 200 ms preceding the onset of the first item of each sound sequence. The resulting sources estimates were transformed to a standard anatomical template (*fsaverage*) with 20,484 vertices using the MNE morphing procedure, and averaged across subjects.

## Multivariate analyses

Data was smoothed with a 100 ms sliding window and, instead of working with the 306 sensors (102 magnetometers and 206 gradiometers), we projected the spherical sources of signal onto the magnetometers using MNE epochs method *epochs.as_type('mag',mode='accurate')*.

### Time-resolved multivariate decoding of brain responses to standard and deviant sounds

The goal of multivariate of time-resolved decoding analyses was to predict from single-trial brain activity ($X$) a specific categorical variable ($y$), namely if the trial corresponded to the presentation of a deviant sound or not. These analyses were performed using MNE-python function *GeneralizingEstimator* from version 0.23.0 (*Gramfort et al., 2013*) and with Scikit-learn package version 1.1.1 (*Pedregosa et al., 2011*). Prior to model fitting, channels were z-scored across trials for every time point. The estimator was fitted on each participant separately, across all MEG sensors using the parameters set to their default values provided by the Scikit-Learn package (*Pedregosa et al., 2011*).

### Cross-validation

One run was dedicated to each version of the sequence (7 sequence types × 2 versions [starting with A or starting with B]=14 runs). To build the training set, we randomly picked one run for each

sequence, irrespectively of the sequence version. We trained the decoder on all deviant trials of the 7 sequences and on standard trials (non-deviant trials from the test phase) that were matched to sequence-specific deviants in ordinal position. We then tested this decoder on the remaining 7 blocks, determining its performance for the 7 sequences separately. The training and the testing sets were then inverted, resulting in a twofold cross-validation. This procedure avoided any confound with item identity, as the sounds A and B were swapped in the cross-validation folds.

## Generalization across time

To access the temporal organization of the neural representations, we computed the generalization-across-time matrices (*King and Dehaene, 2014*). These matrices represent the decoding score of an estimator trained at time $t$ (training time on the vertical axis) and tested with data from another time $t'$ (testing time on the horizontal axis).

## Statistical analyses

Temporal, spatiotemporal, and temporal-temporal cluster-based permutation tests were computed on the time windows of interest (0–350 ms for habituation and standard items and 0–600 ms for deviants) using *stats.permutation_cluster_1samp_test* from MNE python package. To obtain the significance results presented in *Figure 8*, we ran the temporal permutation test on the difference between the predictions of the deviancy decoder for standard and deviant sounds for a time window from 0 to 600 ms after deviant onset. To compute spatiotemporal clusters, we provided the function with an adjacency matrix from *mne.channels.find_ch_connectivity*.

# Acknowledgements

This work was supported by INSERM (Institut National de la Santé et de la Recherche Médicale), CEA (Commissariat à l'Energie Atomique et aux Energies Alternatives), Collège de France, the Bettencourt-Schueller foundation, and a European Research Council ERC grant 'NeuroSyntax' to S.D. We gratefully acknowledge extensive discussions with Ghislaine Dehaene-Lambertz, Christophe Pallier, Maxime, Maheu, Lucas Benjamin, Mathias Sablé-Meyer. We thank all the volunteers for their participation. We are grateful to the UNIACT team for their help in recruiting subjects and in data acquisition.

# Additional information

## Funding

| Funder | Grant reference number | Author |
|---|---|---|
| European Research Council | NeuroSyntax - Grant agreement ID: 695403 | Stanislas Dehaene |

The funders had no role in study design, data collection and interpretation, or the decision to submit the work for publication.

## Author contributions

Fosca Al Roumi, Samuel Planton, Conceptualization, Data curation, Formal analysis, Investigation, Visualization, Methodology, Writing – original draft, Writing – review and editing; Liping Wang, Conceptualization, Writing – review and editing; Stanislas Dehaene, Conceptualization, Resources, Formal analysis, Supervision, Funding acquisition, Investigation, Methodology, Writing – review and editing

## Author ORCIDs

Fosca Al Roumi ⓘ https://orcid.org/0000-0001-9590-080X
Samuel Planton ⓘ https://orcid.org/0000-0001-8588-7146
Liping Wang ⓘ https://orcid.org/0000-0003-2038-0234
Stanislas Dehaene ⓘ http://orcid.org/0000-0002-7418-8275

### Ethics

In compliance with institutional guidelines, all subjects gave written informed consent prior to enrollment. They received 90€ as compensation. The experiments were approved by the national ethical committees (CPP 100049 and 100050).

### Decision letter and Author response

Decision letter https://doi.org/10.7554/eLife.84376.sa1
Author response https://doi.org/10.7554/eLife.84376.sa2

## Additional files

### Supplementary files

• Supplementary file 1. Selected sequences. The first column indicates the list of the different 16-item sequences used in the magneto-encephalography (MEG) and fMRI experiments. *Sequences used only in the fMRI experiment. The second column provides the sequence description obtained from the LoT. The third column is its verbal description, meant to ease the understanding of the formal expression.

• Supplementary file 2. fMRI complexity effect on standard trials (voxel-wise p<0.001, uncorrected; cluster-wise p<0.05, FDR corrected).

• Supplementary file 3. fMRI complexity effect on deviant trials (voxel-wise p<0.001, uncorrected; cluster-wise p<0.05, FDR corrected).

• MDAR checklist

### Data availability

The raw MEG and fMRI raw data are available via OpenNeuro (https://openneuro.org/datasets/ds004483/versions/1.0.0 and https://openneuro.org/datasets/ds004482/versions/1.0.0).

The following datasets were generated:

| Author(s) | Year | Dataset title | Dataset URL | Database and Identifier |
|---|---|---|---|---|
| Planton S, Roumi FA, Wang L, Dehaene S | 2023 | ABSeqMEG | https://doi.org/10.18112/openneuro.ds004483.v1.0.0 | OpenNeuro, 10.18112/openneuro.ds004483.v1.0.0 |
| Planton S, Roumi FA, Wang L, Dehaene S | 2023 | ABSeqfMRI | https://doi.org/10.18112/openneuro.ds004482.v1.0.0 | OpenNeuro, 10.18112/openneuro.ds004482.v1.0.0 |

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
