## [Editor Report]

This article brings to bear an important set of behavioral methods and neural data reporting that activity in numerous cortical regions robustly covaries with the complexity of tone sequences encoded in memory. The study provides convincing evidence that humans store these sequences based on representations related to the so-called language of thought.

---

## [Decision Letter]

**Decision letter after peer review:**

Thank you for submitting your article "Compression of binary sound sequences in human working memory" for consideration by *eLife*. Your article has been reviewed by 2 peer reviewers, and the evaluation has been overseen by a Reviewing Editor and Barbara Shinn-Cunningham as the Senior Editor. The following individuals involved in review of your submission have agreed to reveal their identity: Thomas Christophel (co-reviewed with Joana Seabra).

Essential revisions:

A revision should thoroughly address the many points raised by both reviewers, but let us editorially highlight especially the following requests/concerns:

– As outlined by Reviewer 1, a core issue of the manuscript in its present form is the missing formal description of the LoT model and alternatives, inconsistencies in the model comparisons, and no clear argumentation that would allow the reader to understand the selection of the alternative model. Similar to a recent paper by similar authors (Planton et al., 2021 PLOS Computational Biology), an explicit model comparison analysis would allow a much stronger conclusion. Also, these analyses would provide a more extensive evidence base for the favored LoT model.

– We deem it desirable, if not required, to make the analysis pipeline and the raw data publicly available to investigate the pipeline in detail and reproduce the results (provided hyperlinks are not functional).

– As outlined by Reviewer 2, it is unclear whether the authors have successfully "characterize[d] the mental representation that humans utilize to encode binary sequences of sounds in working memory and detect occasional deviants" as they set out to do. While it is possible that the regions found "constitute the minimal network needed to track sequences" we find the evidence insufficient to assert this.

Further analyses might be needed to elucidate the temporal evolution of the neural signals over its repeated presentation over habituation and test, as well as the experiment overall. Further research might be needed to study persistent activity while minimizing influences from perceptual processing and could even distinguish the representation of individual sequences from each other using pattern analysis.

– Please reconsider the terminology used to describe the proposed model, as argued convincingly and exemplified by Reviewer 2.

*Reviewer #1 (Recommendations for the authors):*

Weaknesses:

Inconsistencies in the analysis across datasets, for example, a direct comparison of the alternative model with the LoT on the behavioral data (but no figure for the alternative model effect), no analysis including the alternative model for the fMRI data, and providing the linear regression analysis for the MEG data that includes the predictor related to the alternative model only as a secondary analysis is unfortunate. It weakens the main conclusion of the manuscript. The manuscript could establish more robust conclusions when, for example, the or multiple alternative models would be treated equally (e.g., figures for the behavioral data). For the fMRI analysis, one could compare the models with an overlap analysis (similar to in Figure 4) and compare effect sizes from different regions. For the MEG data, one could focus on the analysis that considers both models, as one could expect that the model has a higher model fit and provide extensive descriptions for both effects.

Also, a critical case for the model is the 0-correlation of behavior and LoT for the Alternate condition. The authors choose not to include a correlation between behavior in the bracket task and the alternative account. Moreover, no critical discussion of this inconsistency is included.

Moreover, recent developments indicate selective implementation of representations depending on the complexity of the task (e.g., see Boominathan, L., and Pitkow, X. (2022). Phase transitions in when feedback is useful. Advances in Neural Information Processing Systems, 35, 10849-10861.) the investigation of an interaction of the complexity parameter and the transitional probabilities would be highly intriguing.

Further points:

It is highly recommended to make the analysis pipeline and the raw data publicly available to investigate the pipeline in detail and reproduce the results (provided hyperlinks are not functional).

Figure 1.: For a clear understanding for the reader, one could provide the tree structure for the experimental conditions reflecting the LoT complexity

Figure 2 Please present the p-value and the r in bold letters. The current presentation suggests higher importance for the statistics than the effect size estimate.

Results

195-199. As indicated before, the low correlation between bracketing behavior and the LoT model simulations is low. The following section is provided but unclear from reading the previous sections. Please provide more context here and elaborate in the discussion.

"For the latter, a minor departure from the proposed encoding was found, as the shortest LoT representation (i.e. [+0]^16) can be paraphrased as "16 alternations", while the participants' parses corresponded to "8 AB pairs". The latter encoding, however, only has a marginally larger complexity, so this deviation should not affect subsequent results. "

211-219: Be precise on the formulation of the linear mixed effect model. As the random effect structure can vary drastically, it is essential to present the complete form of the model, especially when reanalyzing it with and without the subject random effect. Also, no indication of the inclusion of random slopes or other random effects like stimuli is included.

220-230: Statistical learning as an alternative explanation. The critical test needs to be included. Has the stats learning or the LoT complexity predictor the higher model fit? Also, why not use a metric that penalizes for the number of predictors in the model (e.g., Akaike Information Criterion). Also, please provide the correlation of the two predictors from the models. A similarity estimation is essential to interpret the findings.

341-345: Why use a quadratic trend, and what is the benefit of making the statistical model more complex here? Please elaborate

364" "for loops" postulated in our language"; Please elaborate.

fMRI part:

Comparison analysis based on the alternative model (based on transitional probabilities) needs to be included. This analysis would allow us to investigate which regions contribute to the added behavioral explainability found in the behavioral analysis.

MEG part:

Unclear why the quadratic trend used for the fMRI analysis is not considered here. Please provide an argumentation.

Alternative model, based on transitional probabilities. Figure S4 provides the complexity effects based on a regression model, including the transitional probabilities measure. For clarity, please provide the corrected version in Figure 6 alongside the effect of the alternative model. Again model comparison or if both show considerable effects, one could investigate the interaction of complexity and transitional probabilities. Similarly, use both parameters or the interaction for the classifier data presented in Figure 7.

L 534: Please provide an argument here, why the leave-one-out was used here. Alternatively, one could use a 5-fold cross-validation that is more efficient to estimate and less problematic when it comes to overfitting. I expect this analysis was implemented because the analysis presented in Figure 7 is based only on the unseen dataset.

L 628 It is not clear what statistical test established cross decoding, as the cluster test provided cannot differentiate between time windows (see Sassenhagen, J., and Draschkow, D. (2019). Cluster‐based permutation tests of MEG/EEG data do not establish significance of effect latency or location. Psychophysiology, 56(6), e13335.).

L 763 "fMRI, MEG and behavioral responses to deviants vary linearly with complexity, while model-related fMRI activations vary as an inverted U function of complexity". This result was not tested for, e.g., the MEG data. As described, no non-linear relationships have been investigated in the study.

Subjects:

Please justify the number of participants participating in the experiment using a Power analysis. If no a-priory analysis was implemented, provide a post hoc analysis. This is especially important because it was previously shown that low power could result in false positive inflation for fMRI data (e.g., Yarkoni, T. (2009). Big correlations in little studies: Inflated fMRI correlations reflect low statistical power-Commentary on Vul et al.(2009). Perspectives on psychological science, 4(3), 294-298.)

L 924-929: Please provide the threshold r used for the automatic detection. Also, explain why the number of components was fixed to 1 for cardio and 2 for eye-movement artifacts.

*Reviewer #2 (Recommendations for the authors):*

Before we share our evaluations and concerns, we would like to emphasize that our primary expertise is within working memory research using fMRI and behavioral methods. We hope that our comments will prove to be helpful to the authors.

Is this 'in human working memory'

It is almost impossible to design an experiment that fully precludes some use of working memory. This means that to argue that a process is selectively contained within working memory requires the experimenter to rule out influences of perceptual and long-term memory processes.

Behaviorally, this would mean designing a task where information has to be retained over time, and that discourages the use of long-term memory. Instead, the complexity of the stimuli used (which are likely to exceed working memory capacity) and the intense ongoing stimulation (operating in part as a distractor to working memory processing) seems to encourage the use of long-term memory. Consistently, subjects need to be 'habituated' to the to-be-memorized sequence about ten times, making an influence of early forms of long-term memory very likely.

More importantly, when a neural signal is to be attributed to working memory processing this typically requires some claim of delay period activity (i.e. some form of persistent activity). Here, there is little evidence of differential delay-period activity that is different for different task conditions. Instead, MEG data is most reliably affected by sequence complexity between 100 and 200 ms after the onset of an individual stimulus and complexity specific responses only extend beyond this when a violation of the sequence is detected. The fMRI data does little to alleviate this concern and evaluating this question is made harder by the fact that we could not directly link the time-resolved fMRI data time courses (Figure 3B) with the experimental timeline presented in the methods section.

Despite this, the authors seem to argue that the signals found are a kind of load-like signal that is indicative of the storage of the compressed sequences. For this, it is worth noting that whether load-signals are indicative of an underlying representation of memorized content is questionable (Emrich et al., 2013) and that we know of no instance where the load signals reverse like it does here in deviant trials. The authors, however, argue that the successful quadratic fits of the neural complexity signals are akin to ceiling-in-load effects found in the working memory literature (Vogel and Machizawa, 2004) where neural load-signals reach a ceiling for loads that exceed working memory capacity. This ceiling-in-load effect, however, relies on this tight link between neural and behavioral data evidence which is not presented. Furthermore, the 'quadratic' effect reported here seems to be expressed as data for both the most and least complex sequences diverging from the linear trend. This simply might indicate that the precise complexity of extremely simple and highly complex sequences is less precisely captured by the compression model.

In the introduction the authors, however, also highlight a potentially more specific interpretation of their signals. "[…] the internal model of the sequence, as described by the postulated LoT, would be encoded by prefrontal regions of the brain, and would send anticipation signals to auditory areas, where they would be subtracted from incoming signals. As a consequence, we predict a reciprocal effect of LoT on the brain signals during habituation and during deviancy. In the habituation part of the experiment, lower amplitude response signals should be observed for sequences of low complexity – and conversely, during low complexity sequences, we expect top-down predictions to be stronger and therefore deviants to elicit larger responses, than for complex, hard to predict sequences." To us, this prediction-error-like interpretation seems to be more in line with the data presented. We believe that the time-courses presented in MEG closely match both early and late mismatch signals (mismatch activity and P300). The fMRI data is to the extent we can judge consistent with this interpretation, as is the apparent decline in complexity modulation from habituation to deviant free test trials and the inversion when deviance is detected. The authors argue, in this regard, that the complexity effect is not solely explained by a 'surprise' model. But at least some of the variance in question is explained by surprise. This leaves us to ask whether a better surprise or mismatch model could be found than the one used to capture the results fully (the current implementation of which should be fully explained in the methods). In the end, the more complex a sequence, the less predictable it is and the more surprising its individual elements are when initially presented.

Thus, it is unclear whether the authors have successfully "characterize[d] the mental representation that humans utilize to encode binary sequences of sounds in working memory and detect occasional deviants" as they set out to do. While it is possible that the regions found "constitute the minimal network needed to track sequences" we find the evidence insufficient to assert this.

Further analyses might help elucidate the temporal evolution of the neural signals over its repeated presentation over habituation and test, as well as the experiment overall. Further research might be needed to study persistent activity while minimizing influences from perceptual processing and could even distinguish the representation of individual sequences from each other using pattern analysis. Sequence complexity might be seen as a confound in such research as it leads to prediction-error-like or load-like response that can be independent of the representation of the individual sequence.

The essence of our concerns [has been covered above]. In short, we are convinced that the evidence speaks to a prediction-error-like signal that covaries with the complexity, and not a signal directly related to working memory retention. Our first major recommendation is therefore to put substantially more emphasis on this alternative interpretation of the results.

Arguing for a working memory-specific signal would – in our view – require directly establishing the presence of some form of persistent activity that is linked to the current results. Because of the rapid presentation of the sequences and the absence of a delay, we see no avenue to pursue this in the current fMRI data. Even for the EEG data, one would have to look at the brief time periods between individual habituations and see whether the recorded data carries information about the presented sequences. Even in this case, one would need to find a way to carefully distinguish these results from the prediction-area-like signals found so far. It is possible the authors intended to establish a connection to working memory in a way that we did not comprehend. In this case, this point requires clarification.

It would be of interest to further understand the role of long-term memory in this task. For this, it would be helpful to investigate how the complexity effect changes over the course of the experiment. Comparing the first and second (inverted) presentation of the sequence could establish whether the complexity effect is mediated by long-term memory.

Finally, we would suggest reconsidering the terminology used to describe the proposed model. On first read, we got distracted by questioning what the authors meant by 'language of thought' and how it related to the experimentation presented (e.g. "The present results therefore support the hypothesis that the human brain hosts multiple internal languages"). The terminology currently used implies a direct link to the subject's subjective experience of thought and while we all want to know how the words (and 'words') we 'think' come to be, we do not see how the data presented speak to that question, directly.

---

## [Author Response]

Essential revisions:A revision should thoroughly address the many points raised by both reviewers, but let us editorially highlight especially the following requests/concerns:– As outlined by Reviewer 1, a core issue of the manuscript in its present form is the missing formal description of the LoT model and alternatives, inconsistencies in the model comparisons, and no clear argumentation that would allow the reader to understand the selection of the alternative model. Similar to a recent paper by similar authors (Planton et al., 2021 PLOS Computational Biology), an explicit model comparison analysis would allow a much stronger conclusion. Also, these analyses would provide a more extensive evidence base for the favored LoT model.

We wish to thank Reviewer 1 for their request, which have led to a significant enrichment of the manuscript:

We added a formal description of the LoT in the subsection *The Language of Thought for binary sequences* in the *Results section*. We have added the formal LoT description of the selected sequences in Table S1 as well as their verbal description.Furthermore, we performed a model comparison similar to the one done in (Planton et al., 2021 PLOS Computational Biology). This new analysis beautifully replicates our previous results, and provides more support for the proposed model. This analysis is now included in Figure 2 and in the *Behavioral data* subsection of the *Results section*.

– We deem it desirable, if not required, to make the analysis pipeline and the raw data publicly available to investigate the pipeline in detail and reproduce the results (provided hyperlinks are not functional).

The pipeline and data are now available online.

Data: https://openneuro.org/datasets/ds004483/versions/1.0.0 and https://openneuro.org/datasets/ds004482/versions/1.0.0

Analysis and stimulation scripts: https://github.com/sam-planton/ABseqfMRI_scripts and https://github.com/Fosca/Binary_Sequences

– As outlined by Reviewer 2, it is unclear whether the authors have successfully "characterize[d] the mental representation that humans utilize to encode binary sequences of sounds in working memory and detect occasional deviants" as they set out to do. While it is possible that the regions found "constitute the minimal network needed to track sequences" we find the evidence insufficient to assert this.

We have removed the term “working memory” from the title. The main claim of this work is not about the type of memory in which the compressed information is stored. We thus have changed the title of the article to Brain imaging evidence for compression of binary sound sequences in human memory. It seems that the notion of Working Memory (WM) is defined differently for researchers from different fields. Here, we consider Baddeley’s 1986’s definition of working memory as “a system for the temporary holding and manipulation of information during the performance of a range of cognitive tasks such as [language] comprehension, learning and reasoning”. WM can then be required in goal directed behaviours, in planning and executive control. Here, the sequence compressed description is stored in a temporary memory store that follows the above definition of WM. We thus have kept in the manuscript the term “working memory” when it was used to refer to these properties.

We also removed the sentence “constitute the minimal network…”, since indeed we have no evidence that this network is minimal.

Further analyses might be needed to elucidate the temporal evolution of the neural signals over its repeated presentation over habituation and test, as well as the experiment overall. Further research might be needed to study persistent activity while minimizing influences from perceptual processing and could even distinguish the representation of individual sequences from each other using pattern analysis.

We agree that the suggested analyses, applied to a delay period without stimuli, could be useful to investigate in depth the type of memory that is involved in this experiment. The present experiments, however, are sufficient to our goal, which is to investigate how the structure of a binary melody affects its encoding in the human brain, and to test the predictions of a specific model (language of thought) against others.

– Please reconsider the terminology used to describe the proposed model, as argued convincingly and exemplified by Reviewer 2.

We would like to keep the notion of "language of thought”, which was introduced by Fodor in 1975 and that we present in the first paragraph of the Introduction. We thank the Reviewers for noting that this concept was not clearly defined and not detailed enough. We have now added a subsection on the “Language of Thought for binary sequences” in the *Results section* that provides additional information on the model.

Reviewer #1 (Recommendations for the authors):Weaknesses:Inconsistencies in the analysis across datasets, for example, a direct comparison of the alternative model with the LoT on the behavioral data (but no figure for the alternative model effect), no analysis including the alternative model for the fMRI data, and providing the linear regression analysis for the MEG data that includes the predictor related to the alternative model only as a secondary analysis is unfortunate. It weakens the main conclusion of the manuscript.

We thank the Reviewer and are sorry if our manuscript appeared to suffer from inconsistencies. We have endeavored to address the multiple points raised by the referee.

Behavior. We have added an entire section and a new figure panel in Figure 2, to perform model comparison on behavioral data. We now contrast our LoT model with the other 5 main models of sequence encoding (see below). LoT complexity once again appears as the best predictor of behavior, as previously shown by our work (Planton et al. 2020)Once this point is settled, we no longer compare all these models on fMRI and MEG data. We now solely examine LoT complexity while accounting for the possible contributions of statistical transition probabilities. Note that we do not consider that assessing the contribution of statistical processing in addition to the language of thought predictions constitutes a competing (alternative) model. Statistical learning takes place simultaneously to the mechanism we are investigating and therefore accounts for part of the variance. However, statistical learning is a well-known contributor to brain activity, and therefore is important to model as a regressor. This aspect was now clarified by the following paragraph in the Discussion:

“Many levels of sequence processing mechanisms coexist in the human brain (Dehaene et al., 2015) and statistical learning is a well-known contributor to brain activity. Thanks to the high temporal resolution of MEG, we could separate the effects of transition probabilities from those of sequence structure (as also done by, e.g. Bekinschtein et al., 2009; Maheu et al., 2019; Wacongne et al., 2011).”

fMRI and MEG are very different in their temporal dynamics. Given the low temporal resolution of fMRI (TR = 1.81s), characterizing the contribution of surprise in brain activation related to sequence perception, and distinguishing it from the effect of LoT complexity is a more challenging task in fMRI than it is in MEG. Given the shape of the BOLD response, fMRI signals average over much, if not all of the sounds in each sequence, while it is the rapid changes between a sound and the next one (SOA of 250 ms) that are of concern for the surprise regressor. Tracking item-by-item surprise with the low temporal resolution of fMRI would be equivalent to using a smoothed version of our surprise regressor, such as a moving average. However, when we did so, we found that this average tends towards an identical value for many sequences: Pairs, Shrinking, ThreeTwo, CenterMir, Complex, all have the same global transition probabilities (as many repetitions as alternations across the 16 items) and thus the same average surprise value (see Author response image 1). Thus, our fMRI experiment was not designed to identify the contributions of item-by-item surprise.

**Author response image 1. sa2fig1:** 

We have now added the following paragraph in the introduction:“Second, a simpler system of statistical learning, based on transition probabilities, may operate in parallel with LoT predictions (Bekinschtein et al., 2009; Chao et al., 2018; Maheu et al., 2019, 2021; Meyniel et al., 2016; Summerfield and de Lange, 2014). To separate their contributions, we will use multiple predictors in a general linear model of behavior and of MEG signals, whose temporal resolution allows to track individual sequence items (in fMRI, the BOLD response was too slow relative to the sequence rate of 250 ms per item).”

and this sentence at the beginning of the MEG results part:

“The low temporal resolution of fMRI did not permit us to track the brain response to each of the 16 successive sequence items, nor to any local sequence properties such as item-by-item variations in surprise.”

For this reason, in our previous manuscript, we deemed it unreasonable to attempt to extract or to fit fast surprise predictors to fMRI signals evoked by a sequence with 250-ms SOA. Prompted by the referee, we did try. We used one possibility offered in the SPM software which is to use a parametric modulator for surprise, along with the “boxcar” regressor used to model the onsets and durations in each condition. We computed such an alternative GLM that included a surprise parametric modulator for each condition of the experiment. We computed the same contrasts as before, at the group level, to assess how much the LoT complexity effect was affected. As illustrated in Author response image 2, adding the surprise modulator did not affect the complexity contrast in habituation.

The other contrast for the effect of complexity on deviancy was affected a bit more, as shown in Author response image 3 -- but there were much fewer trials with deviants, and the noise was therefore higher:

**Author response image 3. sa2fig3:** 

Overall, we are therefore not confident that it was reasonable to attempt to model both complexity and surprise in the slow fMRI signals. Our design was solely conceived in order to track complexity, which is a sequence-level variable, rather than surprise, which is a much faster item-level variable.

The manuscript could establish more robust conclusions when, for example, the or multiple alternative models would be treated equally (e.g., figures for the behavioral data). For the fMRI analysis, one could compare the models with an overlap analysis (similar to in Figure 4) and compare effect sizes from different regions. For the MEG data, one could focus on the analysis that considers both models, as one could expect that the model has a higher model fit and provide extensive descriptions for both effects.

We have added the suggested model comparison of alternative models on behavioral data (Planton et al., 2021 PLOS Computational Biology). This analysis is now included in Figure 2 and in the Behavioral data subsection of the Results section. It replicates previous behavioral results obtained in *Planton et al., 2021 PLOS Computational Biology,* namely that complexity, as measured by minimal description length is the best predictor of participants’ behaviour.

Based on these behavioral results, we did not proceed further in the investigation of the neural correlates for the alternative models.

Also, a critical case for the model is the 0-correlation of behavior and LoT for the Alternate condition. The authors choose not to include a correlation between behavior in the bracket task and the alternative account.

We now explicitly test the hypothesis that surprise from transition probabilities could explain the bracketing behavior. We computed the correlation between item-by-item surprise and participants’ bracketing. As seen in Author response image 4, for several sequences, those correlations are low or even negative, and thus fail to explain the observed complexity effects.

**Author response image 4. sa2fig4:** 

A new paragraph was introduced in the behavioral results:“It could be suggested that, rather than the structure predicted by the LoT model, participants use transition probabilities to segment a sequence, with rare transitions acting as chunk boundaries. We thus tested the correlations between the group-averaged number-of-brackets vector and the transition-based surprise derived from transition probabilities. There are 15 item-to-item transitions in 16-item sequences, thus brackets before the first and after the last items were excluded from this analysis. Surprise was computed by pooling over all the transitions in a given sequence. Due to lack of variance, a correlation with transition-based surprise was impossible for sequences Repeat (all transitions are 100% predictable repetitions) and Alternate (all transitions are 100% predictable alternations). For other sequences, a positive correlation was found for sequences Quadruplets (r = 0.99, p <.0001), Pairs (r = 0.88, p <.0001), Complex (r = 0.79, p <.0005), Shrinking (r = 0.73, p <.002), ThreeTwo (r = 0.68, p <.006), and CenterMirror (r = 0.64, p <.02). Crucially, however, no positive correlation was found for sequences PairsandAlt.1 (r = -0.48, p = .071) and PairsandAlt.2 (r = -0.58, p <.03). This was due to the fact, in these sequences, that repetitions were the rarest and therefore the most surprising transitions: thus in these sequences, transition probabilities predicted a breaking of the chunks of repeated item, while participants correctly did not do so and placed their brackets at chunk boundaries (see Planton et al., 2021). Thus, although surprise arising from the learning of transition probabilities can partially predict participants’ bracketing behavior in some cases, notably when a sequence is composed of frequent repetitions and rare alternations, this model completely fails in other (e.g., when repetitions are rare), again indicating that higher-level models such as LoT are needed to explain behavior.”

Moreover, no critical discussion of this inconsistency is included.

We have now added a full discussion of this small departure from our theory. The new paragraph now reads:

“Finally, after the experiment, when participants were asked to segment the sequences with brackets, their segmentations closely matched the LoT sequence descriptions. For instance, they segmented the Pairs+Alt.1 sequence as [[AA][BB]][[ABAB]]. The sole exception was the alternate sequence (ABAB…) which was encoded by our theory as 15 alternations, but was bracketed by participants as 8 repetitions of the subsequence AB. This interesting departure from our theory may indicate that, during sequence parsing, participants do not necessarily identify the most compact representation, but wait until a repeated subsequence occurs, and then encode how often it repeats (including nested repetitions). This parsing strategy would yield a minor departure from our proposed encodings. Undoubtedly, there is still room for improvement in our LoT theory, which is highly idealized and does not incorporate real-time parsing constraints. However, an alternative interpretation of the bracketing results is that the visual bracketing task itself may bias the perception of auditory sequences. By making the entire sequence visible at once, including its start and end point, the visual format may have incited subjects to subdivide it into groups of two, while the auditory sequential presentation alone would have encouraged the “15 alternations” encoding. In support of the latter interpretation, we did not find any evidence for grouping by two in the timing of button presses when subjects reproduced the alternate sequence from memory (Al Roumi and Tabbane, unpublished data). More research, with a greater diversity of sequences and tasks, will be needed to understand whether and where the present theory needs to be amended.”

Indeed, in an experiment we are currently running where participants have to reproduce 12-item sequences of spatial positions, there is no effect of chunking by groups of 2 items in the reproduction RTs (the only detectable effect is an increase in RT between the 10th and 11th item, corresponding to participants’ uncertainty about sequence length). See Author response image 5.

**Author response image 5. sa2fig5:** 

Moreover, recent developments indicate selective implementation of representations depending on the complexity of the task (e.g., see Boominathan, L., and Pitkow, X. (2022). Phase transitions in when feedback is useful. Advances in Neural Information Processing Systems, 35, 10849-10861.) the investigation of an interaction of the complexity parameter and the transitional probabilities would be highly intriguing.

We thank the reviewers for this very interesting remark and for pointing to the article. In the current study, the selected sequences do not allow for such an assessment: surprise has mostly the same value for all the sequences. Nonetheless, the goal of the upcoming fMRI article is precisely to determine if the brain regions involved in the representation of sequence structure are distinct or not from the ones of sequence statistical properties and how they interact.

Further points:It is highly recommended to make the analysis pipeline and the raw data publicly available to investigate the pipeline in detail and reproduce the results (provided hyperlinks are not functional).

This has now been fixed, the data and the analysis pipeline are now available online.

Figure 1.: For a clear understanding for the reader, one could provide the tree structure for the experimental conditions reflecting the LoT complexity

For the sake of clarity, we have now added Figure 1—figure supplement 1, that contains the sequence descriptions provided by the Language of Thought and a verbal description of the sequence. We believe this eases the reader’s understanding of the LoT and the sequences formal descriptions.

Figure 2 Please present the p-value and the r in bold letters. The current presentation suggests higher importance for the statistics than the effect size estimate.

We thank the reviewer for this remark. This change has now been done.

Results195-199. As indicated before, the low correlation between bracketing behavior and the LoT model simulations is low. The following section is provided but unclear from reading the previous sections. Please provide more context here and elaborate in the discussion."For the latter, a minor departure from the proposed encoding was found, as the shortest LoT representation (i.e. [+0]^16) can be paraphrased as "16 alternations", while the participants' parses corresponded to "8 AB pairs". The latter encoding, however, only has a marginally larger complexity, so this deviation should not affect subsequent results. "

We hope the various changes to the manuscript that are reported above, in the previous section, now address the Reviewer’s concern.

211-219: Be precise on the formulation of the linear mixed effect model. As the random effect structure can vary drastically, it is essential to present the complete form of the model, especially when reanalyzing it with and without the subject random effect. Also, no indication of the inclusion of random slopes or other random effects like stimuli is included.

We hope the changes to the manuscript and response now address the Reviewer’s concern.

220-230: Statistical learning as an alternative explanation. The critical test needs to be included. Has the stats learning or the LoT complexity predictor the higher model fit? Also, why not use a metric that penalizes for the number of predictors in the model (e.g., Akaike Information Criterion). Also, please provide the correlation of the two predictors from the models. A similarity estimation is essential to interpret the findings.

As explained above, surprise from transitional probabilities is not an alternative model, but a mechanism that takes place in parallel to the one of interest here. There is therefore no strong motivation to determine which predictor has the highest model fit. Nonetheless, Figures S5 and S7 present the regression betas for LoT complexity and surprise from transitional probabilities. By comparing their amplitude, we can estimate their relative contribution to the signal variance.

Regarding this point, several clarifications to the manuscript have now been done. They have been described in the previous responses.

341-345: Why use a quadratic trend, and what is the benefit of making the statistical model more complex here? Please elaborate

This point was now clarified in the Introduction:

“Two subtleties further qualify this overall theoretical picture. First, at the highest level of sequence complexity, sequences cannot be compressed in a simple LoT expression, and therefore we expect the brain areas involved in nested sequence coding to exhibit no further increase in activation, or even a decrease (Vogel and Machizawa, 2004; Wang et al., 2019). We will evaluate the presence of such a non-linear trend by testing a quadratic term for LoT-complexity in addition to a purely linear term in the regression models.”

364" "for loops" postulated in our language"; Please elaborate.

We thank the Reviewer for pointing this unclear terminology. We have now replaced “for loops” by ‘repetitions’ as it is more closely related to the terminology we introduced in the Result section The Language of Thought for binary sequences and in Figure 1—figure supplement 1.

fMRI part:Comparison analysis based on the alternative model (based on transitional probabilities) needs to be included. This analysis would allow us to investigate which regions contribute to the added behavioral explainability found in the behavioral analysis.

We hope that the Reviewer is satisfied with the answer given in the previous section.

MEG part:Unclear why the quadratic trend used for the fMRI analysis is not considered here. Please provide an argumentation.

We hope that the Reviewer is satisfied with the answer given in the previous section.

Alternative model, based on transitional probabilities. Figure S4 provides the complexity effects based on a regression model, including the transitional probabilities measure. For clarity, please provide the corrected version in Figure 6 alongside the effect of the alternative model. Again model comparison or if both show considerable effects, one could investigate the interaction of complexity and transitional probabilities. Similarly, use both parameters or the interaction for the classifier data presented in Figure 7.

We hope that the Reviewer is satisfied with the answer given in the previous section.

L 534: Please provide an argument here, why the leave-one-out was used here. Alternatively, one could use a 5-fold cross-validation that is more efficient to estimate and less problematic when it comes to overfitting. I expect this analysis was implemented because the analysis presented in Figure 7 is based only on the unseen dataset.

In this work, the cross-validation is two-folds, with a leave-one-run-out method. Indeed, for each sequence, there are two runs, each of them investigating the coding of the same sequence structure, but starting with a different tone (e.g. for a given subject, in run number 2 the “Shrinking” sequence starts with tone A, and in run number 9 it starts with tone B). Therefore, if we train a decoder on the data from all the sequences but coming only from one of the two runs, and we find that the decoder generalizes to the other run, then the decoding has to be independently of stimulus identity. Therefore, there is no risk of overfitting.

This is what was written in the Materials and methods:

“One run was dedicated to each version of the sequence (7 sequence types x 2 versions [starting with A or starting with B] = 14 runs). To build the training set, we randomly picked one run for each sequence, irrespectively of the sequence version. We trained the decoder on all deviant trials of the 7 sequences and on standard trials (non-deviant trials from the test phase) that were matched to sequence-specific deviants in ordinal position. We then tested this decoder on the remaining 7 blocks, determining its performance for the 7 sequences separately. The training and the testing sets were then inverted, resulting in a 2-folds cross-validation. This procedure avoided any confound with item identity, as the sounds A and B were swapped in the cross-validation folds.”

L 628 It is not clear what statistical test established cross decoding, as the cluster test provided cannot differentiate between time windows (see Sassenhagen, J., and Draschkow, D. (2019). Cluster‐based permutation tests of MEG/EEG data do not establish significance of effect latency or location. Psychophysiology, 56(6), e13335.).

We thank the reviewer for this precious and pedagogical article that raises an important (and common) mistake that was present in this work. We have now modified accordingly the text when discussing the significance of the results, in order to avoid making claims on the position and time-windows on which the effects are significant.

L 763 "fMRI, MEG and behavioral responses to deviants vary linearly with complexity, while model-related fMRI activations vary as an inverted U function of complexity". This result was not tested for, e.g., the MEG data. As described, no non-linear relationships have been investigated in the study.

We hope the reviewer finds satisfactory the response we previously provided on this point.

Subjects:Please justify the number of participants participating in the experiment using a Power analysis. If no a-priory analysis was implemented, provide a post hoc analysis. This is especially important because it was previously shown that low power could result in false positive inflation for fMRI data (e.g., Yarkoni, T. (2009). Big correlations in little studies: Inflated fMRI correlations reflect low statistical power-Commentary on Vul et al.(2009). Perspectives on psychological science, 4(3), 294-298.)

It is true that the sample size was not justified by a power analysis. We determined the number of participants based on our knowledge of practices for this type of study, while considering the limits of our own resources (in a Covid time period). We are well aware of the debates and concerns about the lack of statistical power of fMRI or behavioral science studies, but we do not think that there is any solution that would not significantly increase the costs of such studies. Increasing the sample size can be just as costly as carrying out an independent pilot study to determine the sample size a priori via power analysis. Such approach also has some flaws; pilot data are underpowered and subject to a lot of variability, thus weakening the capacity to estimate effects sizes and their variance. Besides, in an fMRI study investigating several effects across many conditions and many areas, a power analysis usually requires choosing one or multiple effects of interest with associated regions of interest, which could be seen as a weakness of the approach. A power analysis could attest for a sufficiently powered effect of LoT complexity in the SMA during habituation, while telling nothing about the power of a different contrast (e.g., complexity effect in deviants) in another area (e.g. STG). Power is also dependent on Signal-to-noise ratio, which can vary with coil, sequence choices, etc – the sources of variations are endless, and as a result, we know of no well-accepted procedure for power analysis.

Following the reviewer’s suggestion, we considered the possibility to estimate power using our current data (although we note there is some controversy about the possibility of estimating power in such a post-hoc manner, e.g. Hoenig and Heisey, 2001). However, although there are regular publications on the subject of power analysis in fMRI, these remain rare and are sometimes called into question. Perhaps as a result of these difficulties, some of the tools that were used in the community are now inaccessible or discontinued: Fmripower (Mumford and Nichols, 2008; https://jeanettemumford.org/fmripower/), Neuropowertools (http://www.neuropowertools.org/ Durnez et al., 2016). We were however able to use PowerMap (Joyce and Hayasaka, 2012), and generate a power map for the contrast “Complexity effect in habituation”. Please see the results in author response image 6 (conducted using all the significant positive clusters of the contrast as a mask and using a threshold for the power analysis of p < 0.05 with FDR correction). They indicate that the power is at least of 0.70 for the highest peak of each cluster. It is maximal at the SMA. However, because this method is designed for pilot studies and is highly circular, we do not think that it provides a correct estimate power in a post-hoc manner. We are open to suggestions from the reviewer to perform such a post-hoc power analysis in a reliable manner.

**Author response image 6. sa2fig6:** 

L 924-929: Please provide the threshold r used for the automatic detection. Also, explain why the number of components was fixed to 1 for cardio and 2 for eye-movement artifacts.

The cluster threshold was set automatically based on the F-threshold corresponding to a p-value of 0.05 for the given number of observations (19), namely 4.381.

Reviewer #2 (Recommendations for the authors):Before we share our evaluations and concerns, we would like to emphasize that our primary expertise is within working memory research using fMRI and behavioral methods. We hope that our comments will prove to be helpful to the authors.Is this 'in human working memory'It is almost impossible to design an experiment that fully precludes some use of working memory. This means that to argue that a process is selectively contained within working memory requires the experimenter to rule out influences of perceptual and long-term memory processes.Behaviorally, this would mean designing a task where information has to be retained over time, and that discourages the use of long-term memory. Instead, the complexity of the stimuli used (which are likely to exceed working memory capacity) and the intense ongoing stimulation (operating in part as a distractor to working memory processing) seems to encourage the use of long-term memory. Consistently, subjects need to be 'habituated' to the to-be-memorized sequence about ten times, making an influence of early forms of long-term memory very likely.More importantly, when a neural signal is to be attributed to working memory processing this typically requires some claim of delay period activity (i.e. some form of persistent activity). Here, there is little evidence of differential delay-period activity that is different for different task conditions. Instead, MEG data is most reliably affected by sequence complexity between 100 and 200 ms after the onset of an individual stimulus and complexity specific responses only extend beyond this when a violation of the sequence is detected. The fMRI data does little to alleviate this concern and evaluating this question is made harder by the fact that we could not directly link the time-resolved fMRI data time courses (Figure 3B) with the experimental timeline presented in the methods section.

We are sorry if our paper gave the impression that it is focussed on whether sequence information is in “working memory” as opposed to other forms of memory. This is not our goal, and we have significantly changed the paper by removing the term “working memory” from the title and other places. The main claim of this article is not about the specific type of short-term memory (WM, early form of long-term memory etc) that is recruited to encode the sequences, but about the fact that they are compressed in memory, and that understanding this compression requires the postulation of a specific “language of thought”. This is why we did not design the study to test specific attributes of WM, such as a delay-period activity.

We believe that the notion of Working Memory is defined differently for researchers from different fields. Here, we consider working memory as “a system for the temporary holding and manipulation of information during the performance of a range of cognitive tasks such as [language] comprehension, learning and reasoning”, as defined by Baddeley in 1986. It may be required in goal directed behaviours and more generally in planning and executive control.

In this study, we assume that sequences are compressed in memory according to the description provided by the language. Participants discover this compressed code during habituation and store it in memory. The content of memory will change according to the context: in a different run, another sequence will be compressed and stored in memory to predict next items and detect deviants. Moreover, fMRI results show activations in the fronto-parietal network, that is generally found in WM studies. This is why we had considered that the type of short-term memory that is recruited in this task is working memory, but it is not so central to our paper.

Despite this, the authors seem to argue that the signals found are a kind of load-like signal that is indicative of the storage of the compressed sequences. For this, it is worth noting that whether load-signals are indicative of an underlying representation of memorized content is questionable (Emrich et al., 2013) and that we know of no instance where the load signals reverse like it does here in deviant trials.

We thank the reviewers for this very interesting article. We agree that the current experimental design does not allow to clearly address the questions raised by the reviewer and in the above-mentioned article – but we hope that the referee will agree that our experimental design suffices to our goal, which is to put to a test the specific predictions of the language-of-thought model.

We have started to run a new experiment that should be able to address these issues better, as the trial structure is the following: participants see a sequence, have to hold it in memory (maintenance period) and then have to reproduce it. We will therefore be able to determine, separately for the two main phases of sequence presentation and maintenance, which brain regions represent stimulus identity and which exhibit load-like signals.

In the current work, we assume that, in a similar manner to Vogel et al. 2005, the brain regions that hold in memory the compressed description of the sequence should exhibit an activity proportional to LoT-complexity, i.e. the length of the description that is stored in memory. In this current paradigm, any brain area that is involved in the prediction of the upcoming item should also exhibit this type of activity as it is less costly to decompress a simple expression than a complex one. We agree that the current paradigm does not allow to disentangle between purely predictive mechanisms versus pure memorization – here, those are two sides of the same coin.

The authors, however, argue that the successful quadratic fits of the neural complexity signals are akin to ceiling-in-load effects found in the working memory literature (Vogel and Machizawa, 2004) where neural load-signals reach a ceiling for loads that exceed working memory capacity. This ceiling-in-load effect, however, relies on this tight link between neural and behavioral data evidence which is not presented.

The quadratic trend that we predicted and identified in the data is not attributed to working memory load, but to the fact that, beyond a certain level of complexity (not length), the sequence becomes impossible to compress and to encode in a compact manner using the proposed language-of-thought. We hope that this is clear in the current version of the manuscript, where we tried to re-explain why behavior should not show such a quadratic trend.

The behavioral task involves deviant detection. As complexity increases, and the capacity to encode the sequence becomes increasingly difficult, then collapses, this can only lead to a monotonic reduction in predictions, and therefore reduced deviant detection. Had we tested various such incompressible sequences, we would maybe have found a ceiling effet in behavioral performance: above a given complexity value, participants may start responding at chance. For the chosen sequence complexity values, we indeed do not see this ceiling effect in behaviour.

Furthermore, the 'quadratic' effect reported here seems to be expressed as data for both the most and least complex sequences diverging from the linear trend. This simply might indicate that the precise complexity of extremely simple and highly complex sequences is less precisely captured by the compression model.

As noted in the Discussion section, we found that, while task performance was primarily linearly related to LoT complexity, fMRI activity was not. It is based on this observation that we excluded the interpretation suggested by the reviewer, namely that extremely simple and highly complex sequences is less precisely captured by the compression model, and interpreted the results in terms of collapse of memory capacities. Most complex sequences exceed memory capacity, and such a collapse in memory capacity leads to reduced predictions and therefore reduced violation detection. This explains why behavioral responses to deviants are mainly linearly correlated with complexity, while model-related fMRI activations vary as an inverted U function of complexity.

In the introduction the authors, however, also highlight a potentially more specific interpretation of their signals. "[…] the internal model of the sequence, as described by the postulated LoT, would be encoded by prefrontal regions of the brain, and would send anticipation signals to auditory areas, where they would be subtracted from incoming signals. As a consequence, we predict a reciprocal effect of LoT on the brain signals during habituation and during deviancy. In the habituation part of the experiment, lower amplitude response signals should be observed for sequences of low complexity – and conversely, during low complexity sequences, we expect top-down predictions to be stronger and therefore deviants to elicit larger responses, than for complex, hard to predict sequences." To us, this prediction-error-like interpretation seems to be more in line with the data presented. We believe that the time-courses presented in MEG closely match both early and late mismatch signals (mismatch activity and P300). The fMRI data is to the extent we can judge consistent with this interpretation, as is the apparent decline in complexity modulation from habituation to deviant free test trials and the inversion when deviance is detected.

This very interesting point ties in with the one raised by the reviewers on the previous page. In this work, we do not oppose predictive coding and working memory mechanisms. Instead, we consider that, in the case of the memorization of structured information, as the compressed code that is stored in memory has to be unfolded to predict upcoming events, these two functions have to be jointly recruited, but on different time-scales.

The authors argue, in this regard, that the complexity effect is not solely explained by a 'surprise' model. But at least some of the variance in question is explained by surprise. This leaves us to ask whether a better surprise or mismatch model could be found than the one used to capture the results fully (the current implementation of which should be fully explained in the methods). In the end, the more complex a sequence, the less predictable it is and the more surprising its individual elements are when initially presented.

We apologize for the confusion. In the previous version of the manuscript, in accord with several previous publications, we used the term “surprise” to refer to “surprise arising from transition probabilities”. The reviewer is right, surprise signals are not only due to surprise from transition probabilities but also to observations that are in disagreement with the expectations based on the sequence LoT-description. As the terminology can be misleading, we have now clarified the use of the term ‘surprise’ and replaced it by “transition-based surprise” when its usage was ambiguous.

Thus, it is unclear whether the authors have successfully "characterize[d] the mental representation that humans utilize to encode binary sequences of sounds in working memory and detect occasional deviants" as they set out to do. While it is possible that the regions found "constitute the minimal network needed to track sequences" we find the evidence insufficient to assert this.

Once again, the main claim of this work is not about the type of memory in which the compressed information is stored, and we agree that this experiment was not designed to determine which one is. We thus have changed the title of the article and refocused it. However, we hope that the referee is convinced that our paper (now including explicit model comparisons) strongly supports the view that a pure transition-based view of sequence prediction is not tenable here, and that the concepts of compression and language-of-thought are needed.

Further analyses might help elucidate the temporal evolution of the neural signals over its repeated presentation over habituation and test, as well as the experiment overall. Further research might be needed to study persistent activity while minimizing influences from perceptual processing and could even distinguish the representation of individual sequences from each other using pattern analysis. Sequence complexity might be seen as a confound in such research as it leads to prediction-error-like or load-like response that can be independent of the representation of the individual sequence.

We hope our previous responses address in a convincing manner this point.

The essence of our concerns [has been covered above]. In short, we are convinced that the evidence speaks to a prediction-error-like signal that covaries with the complexity, and not a signal directly related to working memory retention. Our first major recommendation is therefore to put substantially more emphasis on this alternative interpretation of the results.

As explained in the previous responses, we consider that predictive coding and working memory mechanisms coexist and that the memorized information has to be uncompressed to predict upcoming events. These two functions have to be jointly recruited to perform the experiments. As described above, we have made changes to the manuscript to clarify this aspect.

Arguing for a working memory-specific signal would – in our view – require directly establishing the presence of some form of persistent activity that is linked to the current results. Because of the rapid presentation of the sequences and the absence of a delay, we see no avenue to pursue this in the current fMRI data. Even for the EEG data, one would have to look at the brief time periods between individual habituations and see whether the recorded data carries information about the presented sequences. Even in this case, one would need to find a way to carefully distinguish these results from the prediction-area-like signals found so far. It is possible the authors intended to establish a connection to working memory in a way that we did not comprehend. In this case, this point requires clarification.

We hope our previous responses provide a satisfying answer to this point. To summarize, we agree on the fact that the current design is not adequate to determine which precise type of short-term memory is recruited by the brain in this task and it is not the primary goal of this work. This aspect will be investigated in the next experiment.

It would be of interest to further understand the role of long-term memory in this task. For this, it would be helpful to investigate how the complexity effect changes over the course of the experiment. Comparing the first and second (inverted) presentation of the sequence could establish whether the complexity effect is mediated by long-term memory.

We thank the reviewers for this remark. This would definitely be very interesting in a work that focuses on the characterization of the types of memories that are required in this task. It may thus be of great value in our future works.

Finally, we would suggest reconsidering the terminology used to describe the proposed model. On first read, we got distracted by questioning what the authors meant by 'language of thought' and how it related to the experimentation presented (e.g. "The present results therefore support the hypothesis that the human brain hosts multiple internal languages"). The terminology currently used implies a direct link to the subject's subjective experience of thought and while we all want to know how the words (and 'words') we 'think' come to be, we do not see how the data presented speak to that question, directly.

The notion of "language of thought” was introduced by Fodor in 1975 and we present it in the first paragraph of the Introduction. As this was not clear and detailed enough, we have now added a subsection on the “Language of Thought” in the *Results section* that provides additional information on the model.